# Biophysical characterization of Eag chaperones suggests the mechanism of effector transmembrane domain release

Matthew Van Schepdael[1], Iman Asakereh[2], Jake Colautti[3,4], Andrew J. Gierys[1], Kartik Sachar[1], Shehryar Ahmad[3,4,5], Mazdak Khajehpour[2], John C. Whitney [3,4,6] & Gerd Prehna [1] ✉

The type VI secretion system (T6SS) is a dynamic protein nanomachine found in Gram-negative bacteria that secretes toxic effectors into prey-cells. For secretion, effectors require chaperones or adaptors for proper loading onto the T6SS. Effector associated genes (Eags) are a family of T6SS chaperones that stabilize N-terminal transmembrane domains (TMDs) found in thousands of effectors. Eags are essential for secretion and inhibit effector TMDs from prematurely adopting a membrane-penetrative conformation. However, the mechanism of TMD release from its cognate Eag chaperone is unknown. Here, we take a biochemical and biophysical approach to probe the mechanism of TMD binding and dissociation from Eag chaperones. Using steady-state fluorescence, stopped-flow measurements, and bacterial competition assays, we compare the thermodynamics, kinetics, and in vivo chaperone function of wild-type and point variant Eag-TMD complexes. Additionally, we solve an X-ray crystal structure of an Eag-TMD point variant complex that captures an intermediate state of TMD release. Our data reveals the molecular features and specific residue contacts necessary for TMD binding and demonstrates the Eag conformational change required to initiate rapid release of the TMD. Overall, our work details the stability of Eag-TMD complexes and the energetic pathway for the dissociation of effector TMDs from their Eag chaperones.

The ability to secrete proteins into the environment is a central part of bacterial communication and virulence[1]. Secreted proteins often mediate bacterial antagonistic behaviors that shape microbial communities to form complex cooperative and/or competitive networks[2,3]. As such, bacteria have evolved diverse general and specialized secretion mechanisms. These include large membrane-spanning complexes designed specifically for the translocation of proteins into the extracellular environment or even directly into rival cells[4–6]. In Gram-negative bacteria, one such complex is the type VI secretion system

(T6SS). Generally, the T6SS provides bacteria a competitive advantage in an ecological niche[7].

The T6SS is a contractile nanomachine that forcibly injects toxic protein effectors into nearby cells in a contact-dependent manner[2,8,9]. The T6SS is structurally related to the T4 bacteriophage tail[10,11] and consists of a needle, a baseplate, and a membrane-spanning complex. The needle is made of hexameric Hemolysin co-regulated proteins (Hcp) surrounded by a contractile protein sheath that is polymerized starting from the baseplate[10–12]. The membrane-spanning complex

[1]Department of Microbiology, University of Manitoba, Winnipeg, MB, Canada. [2]Department of Chemistry, University of Manitoba, Winnipeg, MB, Canada. [3]Michael DeGroot Institute for Infectious Disease Research, McMaster University, Hamilton, ON, Canada. [4]Department of Biochemistry and Biomedical Sciences, McMaster University, Hamilton, ON, Canada. [5]Temerty Faculty of Medicine, University of Toronto, Toronto, ON, Canada. [6]David Braley Centre for Antibiotic Discovery, McMaster Uuniversity, Hamilton, ON, Canada. ✉e-mail: gerd.prehna@umanitoba.ca

orients the needle for ejection into the environment[13,14]. Within the baseplate, a trimeric valine glycine repeat protein G (VgrG) binds the top of the Hcp needle[12,15]. VgrG forms the tip of the T6SS and serves as a modular platform for effector loading[16,17]. VgrG can also be "sharpened" by a proline-alanine-alanine-arginine (PAAR) domain-containing protein[16,18] or a PIPY domain protein, which both bind VgrG to create a structural spike[19–21]. Both PAAR and PIPY domains can serve to load VgrG with cargo effectors[22–26] and PAAR domains can be covalently linked to an effector to create a specialized or evolved effector[9,27,28]. The T6SS then secretes the entire Hcp-VgrG-PAAR-effector complex. Finally, the effectors inhibit a wide range of essential processes that kill or inhibit the growth of neighbouring prokaryotic and/or eukaryotic cells[6,29].

A critical step in the secretion of many effectors is the selective recruitment or piloting of effectors to the T6SS by accessory proteins that act as chaperones or adaptors. Chaperones stabilize effectors in a "secretion-competent" state and affect protein folding, whereas adaptors typically only aid in the binding of effectors to T6SS machinery. The terms chaperones and adaptors are often used interchangeably in T6SS literature, especially as some chaperones also aid in effector loading[27,30]. Despite the ambiguity, chaperones and adaptors are essential for the proper delivery and toxicity of numerous T6SS effectors[17,21,27,31]. Several T6SS chaperone and adaptor families have been described, including DUF2169, Type VI adaptor proteins (Tap) (DUF4123), and Effector associated genes (Eag) (DUF1795). DUF2169 forms complexes with an effector and a PIPY protein to facilitate loading onto VgrG[19,22,32], Tap adaptors pilot effectors to VgrGs that contain a helix-turn-helix at their C-terminus[17], and Eag chaperones facilitate the binding of prePAAR containing membrane protein effectors to VgrG[27,33,34].

The basic architecture of a prePAAR containing effector is an N-terminal prePAAR motif followed by a transmembrane domain (TMD), a PAAR domain, and a C-terminal toxin domain[27]. prePAAR-containing effectors can be further divided into two classes. Class 1 effectors belong to the Rhs family of proteins and possess a single TMD, while Class 2 effectors possess two TMD regions directly before and after the PAAR domain[27]. The toxin domains of prePAAR effectors are known to localize to the cytoplasm and inhibit competitor cell growth by acting in the cytoplasm of prey cells, such as by ADP-ribosylation of RNA or protein[35–37] or rapid depletion of the electron carriers $NAD^+$ and $NADP^+$[33,38]. The TMDs are thought to facilitate the translocation of the effector across the prey cell cytoplasmic membrane[31]. Prior to secretion, the TMDs are shielded from the aqueous milieu of the producing cell cytoplasm by their associated Eag chaperone. To promote secretion, Eag chaperones bind the effector TMDs and drastically distort their conformation to inhibit TMD folding[27]. This creates a pre-insertion state to both pilot the prePAAR effector to the T6SS and prevent erroneous membrane insertion before secretion. Additionally, the Eag allows the prePAAR motif to complete a zinc binding motif necessary for the effector PAAR domain to recognize VgrG[27,39]. Notably, Eags are specific for their cognate TMDs and exhibit different effector binding modes.

Given that a bound Eag chaperone would inhibit the proposed membrane penetrating function of the effector TMD, the Eag is presumably removed prior to secretion by the T6SS. Consistent with this hypothesis, Eag chaperones are not secreted alongside their associated effectors[22,23,31,40]. However, there is no current data showing how the Eags are separated from their cognate TMDs. To address this mechanistic question, we took a biophysical and biochemical approach to determine how a T6SS effector TMD is released by an Eag chaperone.

Here, we measure the thermodynamic properties and unfolding kinetics of wild-type Eag-TMD complexes compared to several site-specific point variant Eag-TMD complexes. Our data yields the binding energies and dissociation kinetics of Eag-TMD complexes, which

shows that although Eag-TMDs are remarkably stable only a small perturbation of the Eag is required to promote TMD dissociation. Specifically, TMD binding to an Eag depends upon a crucial knob-in-hole packing interaction (large against small side chain packing) and/or a hydrogen bond that, if removed, results in drastically reduced complex stability and rapid effector release. Furthermore, we show that the disruption of these interactions between an Eag and its cognate TMD reduces effector secretion and bacterial fitness, directly correlating our mechanistic findings to biological function. Finally, an Eag-TMD variant X-ray co-crystal complex captures a TMD release intermediate providing key mechanistic details for TMD binding and dissociation from a chaperone. Our work emphasizes the essential role of Eags in TMD folding inhibition and collectively enables us to propose the mechanism for TMD release from Eag chaperones.

## Results

### Effector TMD binding drastically increases Eag chaperone stability

Effector TMDs are highly insoluble and rapidly degrade unless they are bound and stabilized by an Eag chaperone[31,38]. Given this, it is not possible to express and purify the TMDs without also co-expressing the Eag chaperone. The inability to separately purify Eag and TMD proteins precludes the ability to determine the mechanistic details of their interaction using common binding assays[41,42]. To circumvent this problem, we can study the mechanism of effector TMD binding and dissociation by assaying the stability and thermodynamic properties of Eag-TMD complexes compared to unbound Eags. For this study, we use the Eag-TMD complexes SciW-Rhs(1–59) from *Salmonella* Typhimurium and EagT6-Tse6(1–61) from *Pseudomonas aeruginosa*[27]. For simplicity, the complexes will be referred to as SciW-TMD, EagT6-TMD, or Eag-TMD generally. Notably, the Eag chaperones contain tryptophan residues, whereas the TMDs lack tryptophan. As such, by monitoring the change in tryptophan fluorescence during denaturation, we can determine how TMD binding affects the energetics and folding behavior of the Eag chaperones to extract TMD dissociation data.

Using this approach, we first determined the thermostability of the Eag-TMD complexes compared to unbound or apo-Eags. This was done to gauge the stability of the Eag upon TMD binding. When an Eag binds its cognate TMD, there is a large increase in the thermostability of the chaperone. Nano differential scanning fluorimetry (nanoDSF) reveals an increase in melting temperature ($T_m$) of 30 °C for SciW and 12.2 °C for EagT6 (Fig. 1A, B and Supplementary Table 1). This demonstrates that upon TMD binding, the chaperones become as stable as proteins from extremophiles (>65 °C)[43]. The purified protein materials used in these experiments are shown in Supplementary Fig. 1.

Because the TMDs are invisible in nanoDSF due to their lack of tryptophan residues, we also compared our results to thermal melting by differential scanning calorimetry (DSC). This was done to observe if the TMDs have melting transitions distinct from their Eag chaperones. The DSC experiments gave nearly identical results to nanoDSF (Fig. 1C–F). Furthermore, there were no additional melting transitions in the DSC relative to the nanoDSF for the EagT6-TMD complex indicating that the TMD melts cooperatively with the EagT6 chaperone (Fig. 1D). In contrast, the SciW-TMD complex melts in two transitions (Fig. 1E). Two transitions observed by DSC but only one by nanoDSF suggests that the TMD changes conformation while still bound to SciW before the complex melts (first transition, Fig. 1E). However, in agreement with the nanoDSF data the observed changes in the enthalpy of unfolding (ΔH) for both pairs of Eag chaperones indicates that substantial energy is required to dissociate an Eag-TMD complex.

### Effector TMD binding significantly increases the free energy of Eag chaperone unfolding

We next probed Eag chaperone resistance to chemical denaturation. By monitoring Eag and Eag-TMD unfolding by denaturants, we will be

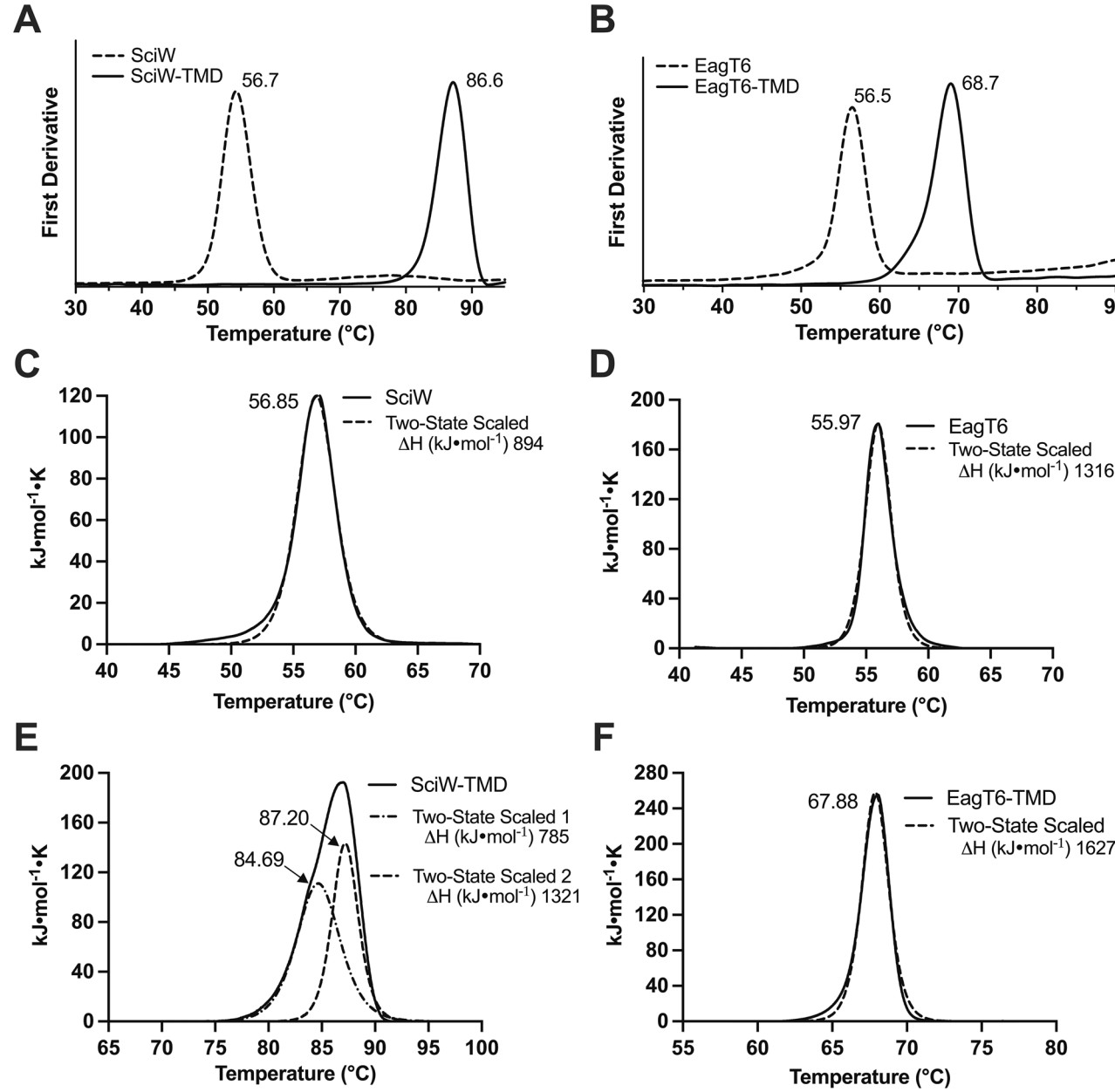

**Fig. 1 | Thermal denaturation of Eag chaperones in complex with effector transmembrane domains.** Thermal denaturation of Eag and Eag-TMD complexes measured by nano differential scanning fluorimetry (nanoDSF). The melting points ($T_m$) of **A** SciW and **B** EagT6 increase to 87 °C and 69 °C, respectively, when bound to their cognate TMDs. Differential scanning calorimetry (DSC) of **C** SciW, **D** EagT6, **E** SciW-TMD, and **F** EagT6-TMD. Data is fit by a two-state model. All transitions are labeled by fitted $T_m$ and enthalpy of unfolding $\Delta H$ values are listed for DSC experiments.

able determine the free-energy ($\Delta G$) of Eag stabilization by the TMD and the binding energy of the Eag-TMD interaction. Specifically, the $\Delta G$ of unfolding measured at various denaturant concentrations is used for extrapolation to 0 M denaturant. This yields native denaturant-independent $\Delta G$s of unfolding and TMD binding energy in a physiologically relevant buffer[44]. Therefore, the data reveals the binding energy that must be overcome during secretion to remove the Eag. We denatured Eag and Eag-TMDs with both urea and guanidine hydrochloride (GdmCl), then monitored steady-state Eag unfolding by tryptophan fluorescence. Shown in Fig. 2A, B and Supplementary Table 1, the Eag-TMD complexes are highly resistant to denaturation by urea. Notably, SciW-TMD could not be unfolded with urea (Fig. 2A). In contrast, the Eag-TMD complexes could be readily unfolded with GdmCl (Fig. 2C, D). The $C_{50}$ (concentration of denaturant at midpoint) values for Fig. 2A–D are summarized in Supplementary Table 1. In

agreement with the thermal denaturation data, the chemical denaturation experiments demonstrate significantly increased Eag stability upon TMD binding.

Now that we have an appropriate denaturant (GdmCl), we can determine the free energy of Eag-TMD unfolding. First, we measured the intrinsic fluorescence spectra of the Eags and the Eag-TMDs as a function of increasing GdmCl concentrations (Supplementary Fig. 2). This was done to experimentally find the wavelength maxima ($\lambda_{max}$) for the unfolding transitions. Measuring the change in $\lambda_{max}$ is a more accurate way to monitor protein unfolding relative to our initial experiments using the Prometheus. The Prometheus compares the intensities measured at the fixed wavelengths of 330 nm and 350 nm, which allows for rapid screening[45,46]. Using the data from Supplementary Fig. 2, the fluorescence emission maximum wavelength $\lambda_{max}$ as a function of denaturant concentration was determined

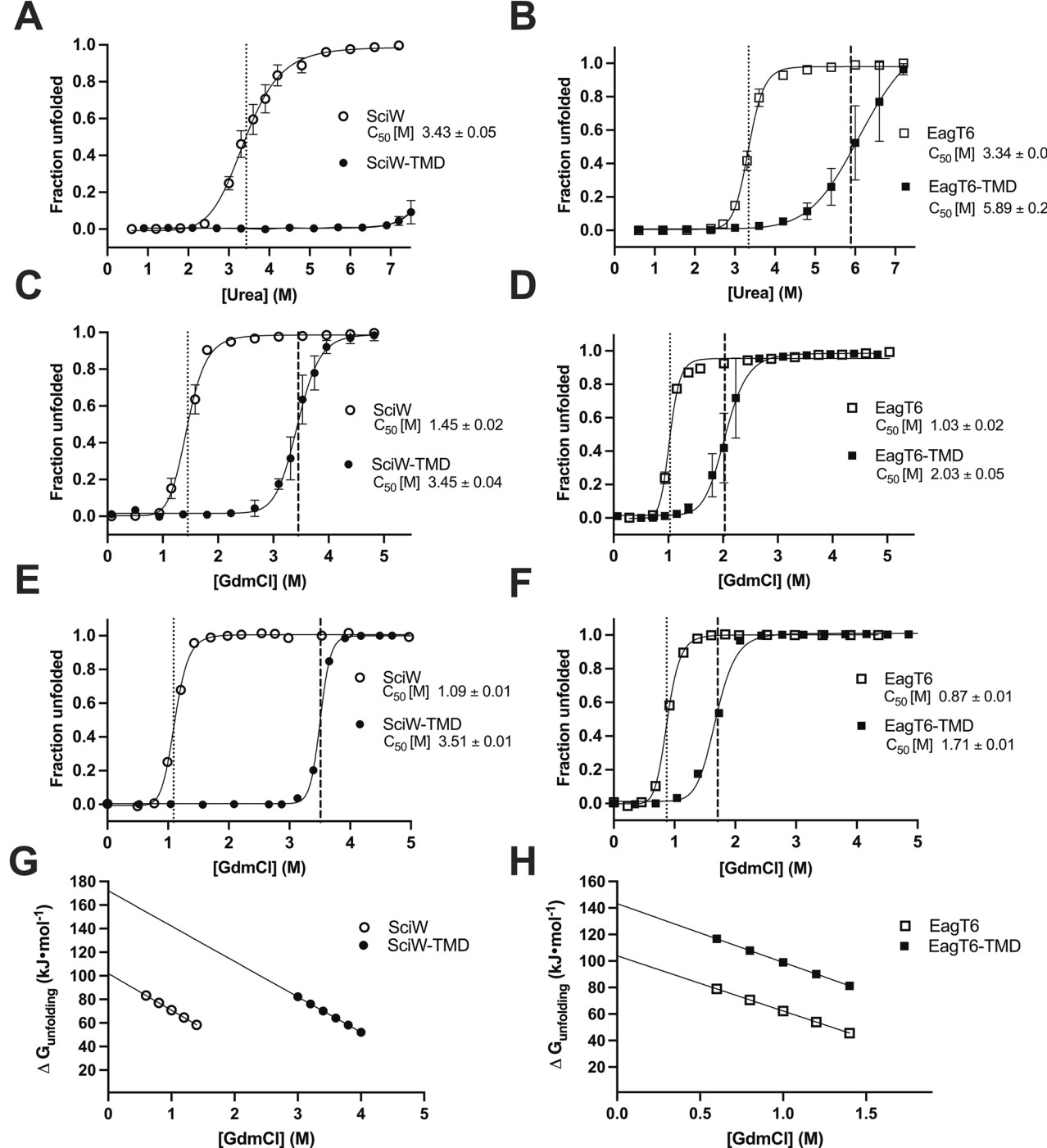

**Fig. 2 | Eag-TMD complexes are resistant to chemical denaturation.** Denaturant induced unfolding profiles measured by nanoDSF for SciW (open circle), EagT6 (open square), SciW-TMD (filled circle), and EagT6-TMD (filled square) as a function of urea (**A**, **B**) or guanidine hydrochloride (**C**, **D**) concentration. Each denaturation was repeated in triplicate, and error bars represent the standard deviation.

**E**, **F** Steady-state unfolding profiles for chaperones as in (**C**, **D**). Data is shown as fraction unfolded. Vertical lines indicate inflection (transition point) at $C_{50}$.
**G**, **H** Linear dependence of $\Delta G_{unfolding}$ on [GdmCl] of Eag and Eag-TMD complex extrapolated to dilute conditions to obtain $\Delta G°$ and the constant of proportionality (m). Results for all parameters are listed in Table 1.

(Supplementary Fig. 3). If we assume that the Eag chaperones unfold through a two-state mechanism (Eq. (1)), the fraction of tryptophan residues in the unfolded state "α" can be related to the measured $\lambda_{max}$[47] to plot the transition (Eqs. (2) and (3)). All equation derivations are detailed in "Methods".

Using Eq. (2), we have plotted α (fraction unfolded) as a function of denaturant concentration for SciW and EagT6 in their apo and TMD-bound forms (Fig. 2E, F and Supplementary Fig. 3). Notably, the experimentally determined $\lambda_{max}$ steady-state fluorescence data is in

close agreement with the fixed 330/350 nm wavelength data (Fig. 2C, D). However, given the slight variation all quantitative thermodynamic and kinetic data will be derived from the experimental $\lambda_{max}$ values.

Next, the data in Fig. 2E, F was used to calculate the change in unfolding free energy ($\Delta G_{unfolding}$) at various denaturant conditions and determine the native $\Delta G°_{unfolding}$ at 0 M denaturant (Eqs. (4)–(6), "Methods")[44]. From fitting the data in Fig. 2E, F to Eq. (5), the values of $\Delta G°_{unfolding}$ and *m* (slope of the linear extrapolation) for the Eag and Eag-TMD complexes were obtained (Table 1). Figure 2G, H shows the

**Table 1 | Unfolding thermodynamic and kinetics of Eag and Eag-TMD complexes**

| Protein | $C_{50}$ [M] | $\Delta G^{\circ}_{unfolding}$ (kJ·mol⁻¹) | $\Delta G^{\circ}_{binding}$ (kJ·mol⁻¹) | $lnk^{\circ}_u$ | $m_{(N-D)}$ | $m'$ |
|---|---|---|---|---|---|---|
| SciW | 1.09 ± 0.01 | 102.0 + 1.3 | −176 | −5.8 ± 0.4 | −31.1 ± 1.1 | 1.62 ± 0.1 |
| SciW-TMD | 3.51 ± 0.01 | 172.2 ± 5.9 | | −10.9 ± 0.9 | −30.0 ± 1.7 | 1.57 ± 0.2 |
| SciW Q58A | 1.20 ± 0.01 | 104.5 ± 1.4 | −142 | −5.0 ± 0.5 | −31.1 ± 1.1 | 1.62 ± 0.1 |
| SciW-TMD Q58A | 3.03 ± 0.02 | 157.5 ± 5.0 | | −6.7 ± 1.0 | −30.0 ± 1.7 | 1.57 ± 0.2 |
| SciW L66A | 1.19 ± 0.01 | 103.5 ± 1.4 | −154 | −5.2 ± 0.5 | −31.1 ± 1.1 | 1.62 ± 0.1 |
| SciW-TMD L66A | 3.19 ± 0.01 | 163.0 ± 5.3 | | −8.9 ± 1.0 | −30.0 ± 1.7 | 1.57 ± 0.2 |
| EagT6 | 0.87 ± 0.01 | 104.0 ± 2.2 | −116 | −1.3 ± 0.2 | −41.7 ± 2.3 | 1.15 ± 0.1 |
| EagT6-TMD | 1.71 ± 0.01 | 143.3 ± 2.0 | | −2.9 ± 0.4 | −44.3 ± 1.7 | 1.14 ± 0.1 |
| EagT6 Q58A | 0.84 ± 0.01 | 103.6 ± 2.1 | −60 | 0.08 ± 0.2 | −41.7 ± 2.3 | 1.15 ± 0.1 |
| EagT6-TMD Q58A | 1.10 ± 0.01 | 115.4 ± 1.3 | | −1.6 ± 0.4 | −44.3 ± 1.7 | 1.14 ± 0.1 |
| EagT6 L66A | 0.58 ± 0.01 | 92.7 ± 1.6 | N/A | −0.045 ± 0.2 | −41.7 ± 2.3 | 1.15 ± 0.1 |
| EagT6-TMD L66A | 0.63 ± 0.01 | 94.9 ± 0.8 | | −0.11 ± 0.4 | −41.7 ± 1.7 | 1.14 ± 0.1 |

Thermodynamic unfolding parameters ($C_{50}$ and $\Delta G^{\circ}_{unfolding}$) and their associated uncertainties were obtained from best global fit of data from Fig. 2 and Fig. 4 to Eqs. (3) and (5), though non-linear least squares regression analysis. $C_{50}$ is the concentration at which 50% of the protein is unfolded and $\Delta G^{\circ}_{unfolding}$ is the free energy of unfolding at 0 M denaturant in physiologically relevant buffer extrapolated in Fig. 4. $\Delta G^{\circ}_{binding}$ is the standard free energy change of the TMD binding the chaperone derived from Eq-7 $m_{(N-D)}$ is the slope from each linear extrapolation and fit of $\Delta G^{\circ}_{unfolding}$ in Eq. (5). Kinetics parameters ($k^{\circ}_{unfolding}$ at 0 M denaturant) and $m'$ and their associated uncertainties were obtained from the best global fit of the data from Fig. 5 through non-linear extrapolation with Eq. (7). Units of $k^{\circ}_{unfolding}$ are 1/s.

plot of $\Delta G_{unfolding}$ as a function of denaturant and the fitted linear extrapolation to 0 M GdmCl[44]. Both Eags have a similar native $\Delta G^{\circ}_{unfolding}$ of ~103 kJ/mole and each gains significant stability as an Eag-TMD complex. When bound to their cognate TMD, the $\Delta G^{\circ}_{unfolding}$ for SciW increases by ~71 kJ/mole and the $\Delta G^{\circ}_{unfolding}$ for EagT6 increases by ~39 kJ/mol. For context, the $\Delta G^{\circ}_{unfolding}$ for lysozyme in a native buffer is ~30 kJ/mol[48].

The effector TMD binding energy ($\Delta G^{\circ}_{binding}$) and association constant ($K_{TMD}$) can then be obtained from comparing Eag $\Delta G^{\circ}_{unfolding}$ to Eag-TMD $\Delta G^{\circ}_{unfolding}$. This calculation requires the following assumptions: (1) the TMD is a small ligand compared to the Eag, (2) TMD binding does not significantly alter the Eag fold, (3) all free TMD in solution results from complex dissociation, (4) the chaperone folds by a 2-state model, and (5) the TMD is assumed to selectively bind the folded form of the Eag with an association constant $K_{TMD}$. Given these, the effect of TMD binding upon the standard unfolding free energy for the chaperone can be estimated using Eq. (7) ("Methods")[49,50]. $\Delta G^{\circ}_{binding}$ is the standard free energy change of TMD binding the chaperone. This directly relates the change in Eag unfolding standard free energy to that of TMD binding at native conditions.

The unfolding data allows us to estimate a $\Delta G^{\circ}_{binding}$ of approximately −176 kJ/mol for the SciW-TMD complex and a $\Delta G^{\circ}_{binding}$ of −116 kJ/mol for the EagT6-TMD complex (Table 1). Antibodies with pico to femto-Molar (fM, 10⁻¹⁵) dissociation constants have a $\Delta G^{\circ}_{binding}$ near −70 kJ/mol[51]. The calculated binding constants for the SciW-TMD and EagT6-TMD interactions are on the order of ~7 × 10⁻³² M and ~5 × 10⁻²¹ M, respectively. Our thermodynamic data (Figs. 1 and 2) clearly shows that the Eag chaperones bind their TMDs extremely tightly and gain significant stability when in complex with a TMD. This in turn raises the question of the biochemical mechanism needed to overcome the extreme binding energy for TMD release.

## Eag chaperone α-helical packing residue mimics act to inhibit TMD folding

Using the thermodynamic properties of the Eag-TMD complexes combined with their known molecular interactions, we can probe the mechanism of TMD dissociation. Co-crystal structures of SciW-TMD and EagT6-TMD show that Eags bind their cognate TMDs by mimicking alpha helical membrane packing[27]. This is accomplished by the Eag dimer folding like a claw around the TMD to create a hydrophobic environment. Additionally, the Eags harbor conserved residues within the pocket of the claw that make hydrophilic bifurcated hydrogen

bonds (H-bonds) and knob-in-hole packing interactions with the TMD (Fig. 3 and Supplementary Fig. 4). The net effect is to inhibit the TMD from making H-bonds and knob-in-hole packing interactions with itself to properly fold and thus remain bound to the Eag.

To determine which Eag molecular interactions are important for TMD binding and complex stability, we performed site-directed mutagenesis on several conserved residues found in both SciW and EagT6 (Fig. 3). The positions of each residue and their specific interactions with the TMD are shown in Fig. 3A, B. Note that due to the domain-swapped architecture of the dimeric Eag chaperones, these contacts are not equivalent in each Eag monomer. The interaction surfaces of each point variant in the opposing chain are also depicted in Supplementary Fig. 4C.

Eag residue I25 (SciW)/I24 (EagT6) was changed to phenylalanine to interfere with the TMD properly packing against the base of the hydrophobic Eag claw (Fig. 3B). The Eag hydrophilic residues S41, Q58, and Q106 (SciW)/Q102 (EagT6) were substituted to alanine because the residues make bifurcated hydrogen bonds to the TMD peptide backbone (Fig. 3B). Side-chain to main-chain bifurcated H-bonds are crucial for transmembrane helices to properly adopt their tertiary structure[52–54]. The conserved Eag residue L66 was substituted to L66A because it is a mimic of "knob-in-hole" packing interactions. Knob-in-hole packing is when a large hydrophobic residue on one TM-helix acts as a knob to fill the hole created by a small residue, such as glycine or alanine on another TM-helix with a GxxxG/A motif[55]. As shown in Fig. 3B Eag L66 acts as a knob to fill a hole with TMD alanine residues. A shorter side-chain (leucine to alanine) removes the knob. Eag L66 was also of interest because it is on the outer part of the Eag claw. Specifically, the loop containing L66 undergoes a conformational change that moves the Eag claw structure over the TMD upon binding[27].

After residue substitution, each apo-Eag variant and Eag-TMD variant was purified and its fold assayed by dynamic light scattering (DLS). As shown in Supplementary Fig. 5 and Supplementary Table 2, all point variants purified similar to wild-type and exhibited the same DLS profile. This demonstrates that like the wild-type Eags each Eag variant is dimeric in solution and not aggregated due to misfolding. Furthermore, all Eag-TMD variants still co-purify bound to their cognate TMD, showing they are all still functional. Given that we know the $\Delta G^{\circ}_{unfolding}$ and $\Delta G^{\circ}_{binding}$ for wild-type Eag-TMDs (Fig. 2), we can now determine how much binding energy is contributed by each Eag residue contact with the TMD and determine mechanistic details of TMD binding and dissociation.

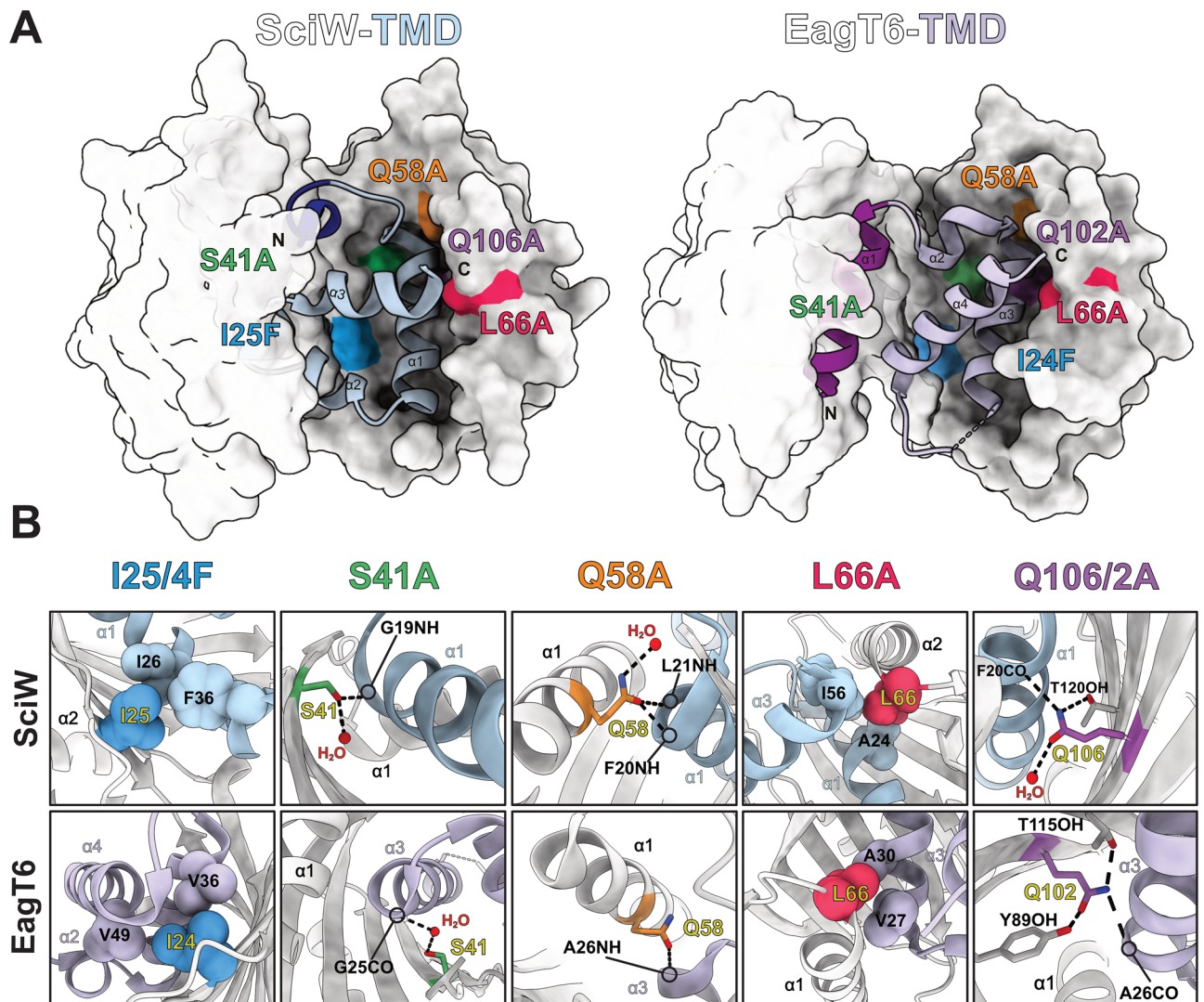

**Fig. 3 | Structural analysis of conserved helical packing interactions between Eag chaperones and TMD helices. A** Structures of SciW-TMD (left) and EagT6-TMD (right) complexes with selected point variants shown at their positions and highlighted by colour. The prePAAR region is shown in dark blue for SciW-TMD (residues 9–13) and dark purple for EagT6-TMD (residues 1–15). **B** Detailed interaction with each SciW (top panel) and EagT6 (bottom panel) point-variant and their associated TMDs, Rhs1, and Tse6, respectively. I25/24 (blue) make hydrophobic packing interactions. S41 (green) and Q58 (orange) both engage in bifurcated H-bonds (dashed-lines) with the TMD. L66 (red) makes knob-in-hole packing interactions with the TMD. Q106/102 (purple) also makes bifurcated H-bonds (dashed-lines) with the TMD. Eag chaperones are shown in light grey and the TMD of Rhs1 in light blue and the TMD1 of Tse6 in light purple. Interactions between the TMD and the second Eag monomer are shown in Supplementary Fig. 4.

## Eag α-helical packing residue mimics contribute significantly to TMD binding

To probe the effect of Eag variants on TMD binding, we screened their resistance to both thermal (Supplementary Fig. 6) and chemical denaturation (Supplementary Fig. 7 and Supplementary Table 3). Both thermal and chemical denaturation show a similar pattern of lower complex stability for the selected point variants. However, the chemical denaturation data was used to assay stability for $\Delta G$ measurements. For SciW-TMD, complex stability was reduced by the residue substitutions but was still significantly more stable than the apo-SciW variants (Supplementary Fig. 7A, C, E, G, I and Supplementary Table 3). When comparing the change in $C_{50}$ between the apo-EagT6 and TMD bound variants, we observe that all substitutions greatly weaken complex stability (Supplementary Fig. 7B, D, F, H, J and Supplementary Table 3). However, EagT6 I24F, L66A, and Q102A appear to have the greatest effect (small or no change in bound to unbound variant $C_{50}$). This shows that EagT6-TMD binding is severely weakened by either the loss of knob-in-hole (L66A) or bifurcated H-bond interactions (Q58A

and Q102A) (Fig. 3 and Supplementary Fig. 4). Additionally, steric hindrance of proper TMD packing by a larger hydrophobic residue (I24F) is also severely disruptive. For the SciW-TMD complex, the loss of bifurcated H-bonds (Q58A) or knob-in-holes (L66A) is substantial but less pronounced (Supplementary Fig. 7 and Supplementary Table 3). Additionally, Q106/2 was important for EagT6-TMD binding but had almost no effect on SciW-TMD binding.

The variants Q58A and L66A represent two distinct modes of α-helix membrane protein packing and affect both SciW-TMD and EagT6-TMD complex stability. Given this, we selected these variants for $\Delta G_{unfolding}$ quantification by intrinsic protein fluorescence spectral analysis. The tryptophan fluorescence wavelength maxima ($\lambda_{max}$) for the Q58A and L66A point variants were determined and used to plot α (fraction unfolded) as a function of denaturant concentration using Eq. (6) (Supplementary Figs. 3 and 8, Fig. 4A–D, and "Methods"). The $\Delta G_{unfolding}$ for the point variants was calculated from fitting α (fraction unfolded) with Eqs. (1)–(6) and $\Delta G°_{binding}$ from Eq. (7), similar to the wild-type. Because the "m" parameter in Eq. (6) only depends upon the

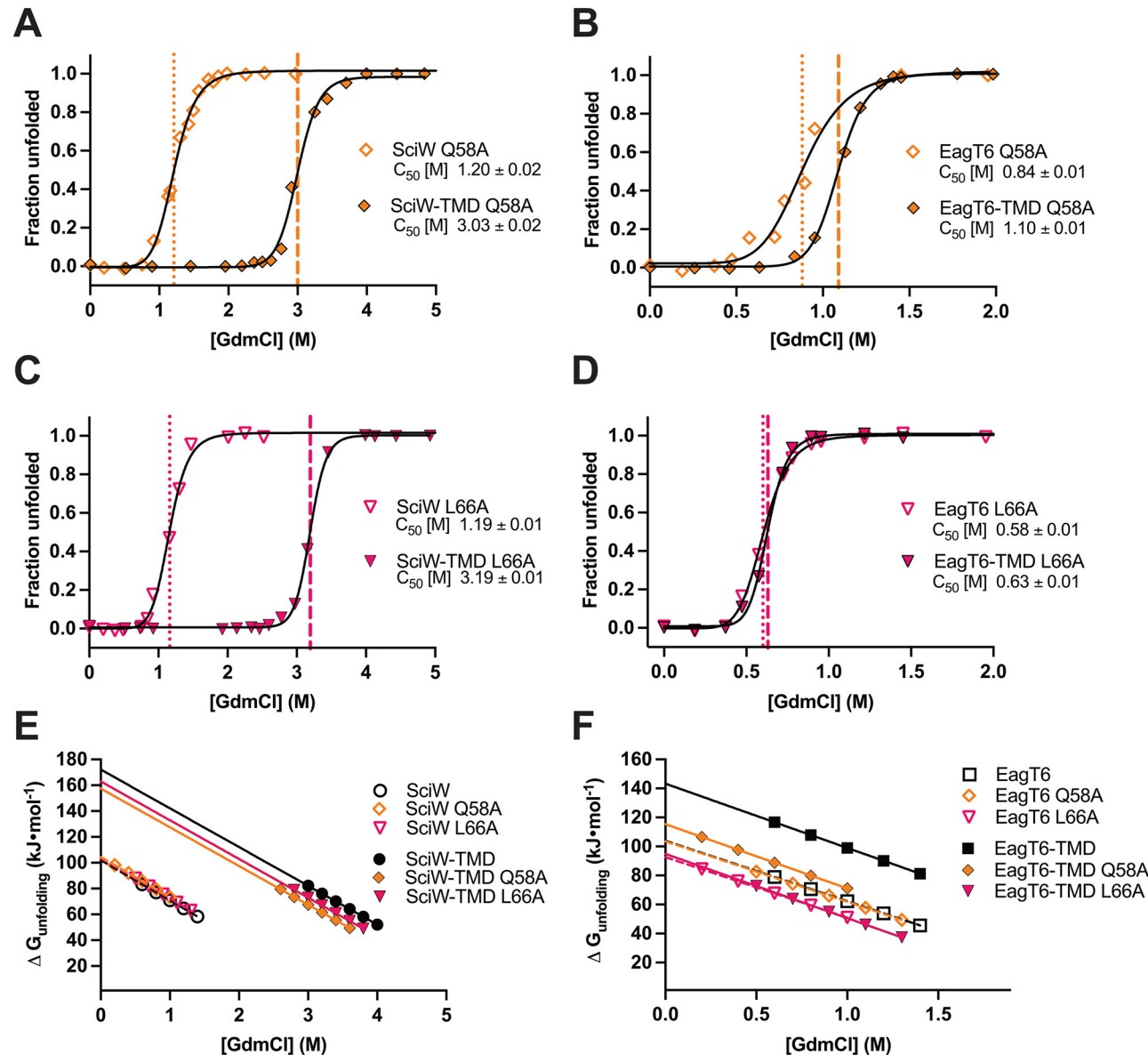

**Fig. 4 | Eag-TMD point variants show reduced complex stability by chemical denaturation.** Denaturant induced unfolding profiles of variant Eag and Eag-TMD complexes. **A**, **B** Q58A variant unfolding profiles. **C**, **D** L66A variants unfolding profiles. **E**, **F** Linear dependence of $\Delta G_{unfolding}$ on [GdmCl] of Eag and Eag-TMD variants extrapolated to dilute conditions to obtain $\Delta G°_{unfolding}$. Eag Q58A variants are plotted as an orange open diamond. Eag-TMD Q58A variants are plotted as an

orange-filled diamond. Eag L66A variants are plotted as a fuchsia open triangle. Eag-TMD L66A variants are plotted as a fuchsia-filled triangle. Wild-type SciW and SciW-TMD complexes are shown as open and black circles, respectively. Wild-type EagT6 and EagT6-TMD complexes are shown as open and black squares, respectively. Wild-type fits are replotted from Fig. 2G, H to compare to variants. The results of all fits are shown in Table 1.

change in surface accessible area upon unfolding[56], the α (fraction unfolded) values were fitted globally, sharing the "m" parameter in these subsets: (1) apo-SciW point-variants, (2) SciW-TMD point-variants, (3) apo-EagT6 point-variants, (4) EagT6-TMD point-variants. The resulting parameters from the fits are listed in Table 1. The dependence of $\Delta G_{unfolding}$ on denaturant concentration with fitted extrapolations to native buffer conditions at 0 M denaturant ($\Delta G°_{unfolding}$) are shown in Fig. 4E, F.

Compared with the wild-type $\Delta G°_{unfolding}$, the point variants Q58A and L66A have little effect on the $\Delta G°_{unfolding}$ for apo-SciW. However, both residue substitutions significantly lowered the $\Delta G°_{unfolding}$ of the SciW-TMD complex, indicating a reduction in TMD binding affinity (Table 1 and Fig. 4A, C, E). The $\Delta G°_{binding}$ values for SciW Q58A and L66A with the TMD are −142 kJ/mol and −154 kJ/mol, respectively, compared to −176 kJ/mol for wild-type. We can conclude that the

bifurcated H-bonds provided by Q58 contribute ~34 kJ/mol to TMD binding, and the knob-in-hole interactions of L66 contribute ~22 kJ/mol. Within experimental error, these are roughly 15–20% of the total SciW-TMD binding energy. This also increases the dissociation constant for Q58A and L66A relative to wild-type by several orders of magnitude to ~$1 \times 10^{-25}$ M and ~$7 \times 10^{-28}$ M, respectively.

For EagT6, the point variants Q58A and L66A have a more pronounced effect on complex stability and TMD binding. The folding free energy of apo-EagT6 is minimally affected by the Q58A substitution, but the stability of EagT6 is reduced by L66A (Table 1 and Fig. 4B, D, F). However, the EagT6 L66A variant chaperone is still properly folded and functional as it co-purifies bound to the TMD (Supplementary Fig. 5). Moreover, we observe drastic reductions in the $\Delta G°_{unfolding}$ for the EagT6-TMD variant complexes showing that Q58 and L66 play critical roles in TMD binding. For the Q58A, the $\Delta G°_{binding}$ to the TMD

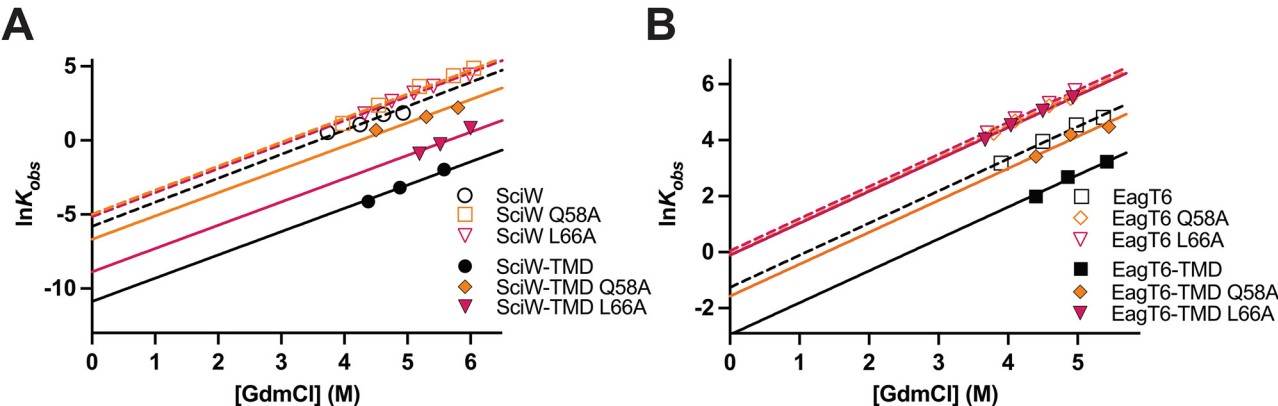

**Fig. 5 | Unfolding kinetics of wild-type and point-variant Eag chaperones.** The observed unfolding rates ($k_{obs}$) of **A** SciW and SciW-TMD and **B** EagT6 and EagT6-TMD as function of GdmCl concentration. Data was linearly extrapolated to 0 M GdmCl to obtain the rate of unfolding in buffer ($k°_u$). Fit parameters ($m_u$) and $\ln k°_u$ for the linearly regression are shown in Table 1. Stopped-flow traces are shown in Supplementary Figs. 9 and 10.

has reduced in magnitude from −116 kJ/mol to −60 kJ/mol. Thus, we can conclude that the bifurcated H-bonds and packing interactions of Q58 with the TMD contribute at least 56 kJ/mol or about 1/2 the binding energy. This increases the dissociation constant by several orders of magnitude to ~3 × 10⁻¹¹ M. For EagT6 L66A, the value of $\Delta G°_{unfolding}$ is similar for the variant apo and TMD bound. This indicates that minimal TMD binding to the EagT6 L66A variant exists under our assay conditions. Our fluorescence experiments were performed under dilute conditions (1 µM chaperone) and in the absence of crowding agents, making it likely that the TMD dissociated prior to the assay. This shows that the binding constant $K_{TMD}$ for the L66A substitution has been reduced to less than $10^6$ M⁻¹ putting an upper limit of ~−35 kJ/mol upon the $\Delta G°_{binding}$. Our data shows that the knob-in-hole packing of residue L66 is crucial for maintaining the EagT6-TMD complex.

### Minor variations to the Eag molecular surface accelerate the rate of TMD release

Our biophysical data thus far has been measured at equilibrium conditions. However, T6SS assembly and firing is a dynamic process that can occur within several seconds[57–59]. Given this, how quickly the TMD can be released prior to secretion is of central importance to the function of the Eag chaperone. To determine the rate of TMD release, we investigated the unfolding kinetics of wild-type and variant Eag chaperones using stopped-flow spectrofluorimetry. Unfolding fluorescence trace data as a function of time for wild-type Eag and Eag-TMD complexes is shown in Supplementary Fig. 9 and point variant traces in Supplementary Fig. 10. The relaxation profiles can be described with a mono-exponential decay and the native unfolding rate constants ($k°_{unfolding}$) calculated using chevron plot analysis by extrapolation to 0 M GdmCl[44] (Fig. 5 and Table 1, and Eqs. (8) and (9), "Methods").

Wild-type SciW and the Q58A and L66A variants have unfolding rate constants ($k°_{unfolding}$) that are indistinguishable within the limits of experimental uncertainty (Fig. 5A and Table 1). The $k°_{unfolding}$ constants reveal that the unfolding/folding lifetime for SciW is on the order of ~5 min. Again, this highlights that the point variants have no effect on the biophysical properties of the SciW chaperone itself. However, both Q58A and L66A significantly increase the rate of unfolding for the SciW-TMD complex (Fig. 5A). The wild-type SciW-TMD complex has a lifetime on the order of ~15 h but is reduced to ~2 h for SciW L66A and ~16 min for SciW Q58A. The disruption of a knob-in-hole (L66) or bifurcated H-bond (Q58) increases the rate of TMD dissociation by 7.5× and 60×, respectively, but only at the cost of 15–20% binding energy (Fig. 4F and Table 1).

The wild-type EagT6 chaperone has a folded state lifetime of ~5 s, which increases to ~18 s when bound to TMD1 of Tse6 (Fig. 5B and

Table 1). The extreme difference in the folded lifetime of apo EagT6 compared to apo SciW was unexpected, as both chaperones have the same $\Delta G°_{unfolding}$ (Table 1) and domain-swapped dimeric structure (Fig. 3)[27,60]. Additionally, we find that both Q58A and L66A increase the rate of unfolding for the apo EagT6 chaperone to ~1 s. Interestingly, this differs from the steady-state data where EagT6 Q58A is indistinguishable from wild-type. However, similar to SciW, both EagT6 Q58A and EagT6 L66A greatly increase the rate of EagT6-TMD complex unfolding. EagT6 Q58A reduces the lifetime of the EagT6-TMD complex to ~5 s. This is a ~4 s difference in lifetime for the Q58A variant as compared to ~13 s for wild-type, demonstrating that the increased TMD dissociation rate is not primarily due to the faster unfolding of the apo-variant. For EagT6 L66A we observe no kinetic difference with the TMD (Fig. 5B and Table 1). Again, this is likely because TMD release by EagT6 L66A is faster than we can measure. In agreement with the steady-state data, the knob-in-hole interactions provided by EagT6 L66 are essential to keep the TMD bound to the Eag.

### The transition state intermediate of TMD release from Eags differs by effector class

The effect of Eag variants upon the unfolding rate ($k°_{unfolding}$) and energy ($\Delta G°_{unfolding}$) also reveals mechanistic details about the TMD dissociation transition state. By applying a "φ-value analysis" to the measured Eag thermodynamic and kinetic parameters we gain insight into the rate-limiting step of the unfolding transition state. The φ-value is the ratio in change of the transition state free energy for unfolding ($\Delta G^{\ddagger}_{unfolding} = -RT(\ln k°_{unfolding})$) to the equilibrium free energy of unfolding ($\Delta G°_{unfolding}$) between wild-type and variants (Eq. (10)). This gives a φ-value between 0 and 1, where 0 represents a dynamic or unstructured transition state and 1 an ordered or structured transition state[61].

The φ-value analysis of the Eag-TMD complexes yields 0.7 ± 0.4 for SciW Q58A, 0.5 ± 0.2 for SciW L66A, and 0.1 ± 0.3 for EagT6 Q58A. A φ-value calculation for EagT6 L66A was not possible because the TMD had already dissociated. The small φ-value for EagT6 Q58A indicates a dynamic or disordered conformation at the unfolding transition state of the EagT6-TMD complex. This could be due to that the TMD is conformationally dynamic at the transition state and/or that the TMD has already been released. Moreover, release of the TMD from EagT6 and the unfolding of EagT6 are uncoupled processes. Given the inability to determine a φ-value for EagT6 L66A a similar disordered transition state is likely. For SciW Q58A and L66A, the φ-values are both 0.5 or higher, which is more characteristic of a structured unfolding transition state. Additionally, the φ-values suggest that the TMD is bound to SciW during the unfolding transition state. This

indicates that the release of the TMD and unfolding of SciW are coupled. Overall, our biophysical characterization shows that the exact mechanistic details and transition state of TMD release differ between SciW (class 1 effector TMDs) and EagT6 (class 2 effector TMDs). First, that disruption of the H-bonds provided by Q58 or Q102, or a knob-in-hole provided by L66 is sufficient to dissociate the TMD from EagT6, but TMD release from SciW requires disruption of more than one contact. Second, that the dissociation of the TMD from SciW involves a stable conformation intermediate of either the TMD, SciW, or both.

### EagT6-TMD stability variants inhibit effector secretion and decrease bacterial fitness

To observe how our biophysical characterization relates to biological activity, we next determined the contribution of our Eag point-variants to the function of the chaperones in vivo using growth competition assays. EagT6-Tse6 was chosen for our model because our point variants have a significantly greater effect on Tse6 TMD release from EagT6 as compared to SciW and the TMD from Rhs1 (Figs. 4 and 5). For example, in EagT6 L66A is sufficient to rapidly release the TMD in vitro. Moreover, EagT6 is known to be essential to T6SS-dependent bacterial competition in *P. aeruginosa*[31,38]. Additionally, the T6SS of *P. aeruginosa* is a highly tractable model system since deletion of the negative T6SS regulator *retS* results in strains with constitutive T6SS activity[17,31,33]. In contrast, the *S.* Typhimurium T6SS requires induction by addition of bile-salts to the media[62]. We were able to show T6SS-dependent killing of *Escherichia coli* by *Salmonella*; however, a *ΔsciW* deletion strain only had an intermediate loss of function phenotype (Supplementary Fig. 11). This is in agreement with past results that showed *ΔsciW* had little effect on *S.* Typhimurium fitness in a mouse model[63]. Given the moderate phenotype of the *ΔsciW* deletion strain and that the SciW point variants do not abrogate binding like EagT6, assays in *Salmonella* were not pursued.

Competition assays were performed with a *P. aeruginosa* donor and an isogenic Tse6-susceptible recipient that lacks Tse6 and the immunity protein Tsi6 (*Δtse6Δtsi6*)[27,31]. The donor strains encode the wild-type EagT6 chaperone (parent), no chaperone (*ΔeagT6*), or one of the EagT6 point variants expressed from their native chromosomal locus. When co-cultured on a solid surface, the wild-type donor readily outcompetes the recipient strain (Fig. 6A). Consistent with our biophysical data, any of the *eagT6* alleles encoding variants of the chaperone display attenuated fitness against the Tse6-susceptible recipient. Specifically, EagT6 L66 strongly contributes to the function of EagT6 in vivo and therefore the proper secretion of Tse6. Furthermore, the donor strain encoding *eagT6(Q58A)* displays intermediate fitness against the recipient strain. This is consistent with our biophysical finding that Q58 contributes less significantly than L66 to the stability of the EagT6-TMD complex. The agreement between these genetic results and our biophysical data suggest that the determinants of complex stability identified in vitro closely reflect the function of the EagT6-TMD complex in *P. aeruginosa*.

Since Tse6 requires EagT6 binding for cytosolic accumulation in *P. aeruginosa*[31], we reasoned that Tse6 levels may be reduced in strains harboring our mutant *eagT6* alleles. Indeed, expression of the EagT6 variants decreased intracellular Tse6 levels (Fig. 6B). Furthermore, the relative abundance of Tse6 in the EagT6 Q58A and L66A variant backgrounds correlates with the contribution of those residues to competitive fitness and in vitro TMD binding energy (Figs. 4–6 and Table 1). Notably, the relative abundance of Tse6 is high for EagT6 Q58A as compared to other the mutants and wild-type (Fig. 6B). Supporting this, EagT6 Q58A has a $\Delta G°_{binding}$ of −60 kJ/mol, which is still indicative of a stable complex. Regardless, in agreement with our biophysical data the loss of TMD binding affinity by Q58A significantly decreases Tse6 secretion (Fig. 6A). Overall, these findings demonstrate that the knob-in-hole packing provided by EagT6 L66 with the TMD of Tse6 is critical for the stability of the EagT6-Tse6 complex in vivo and proper secretion of the effector.

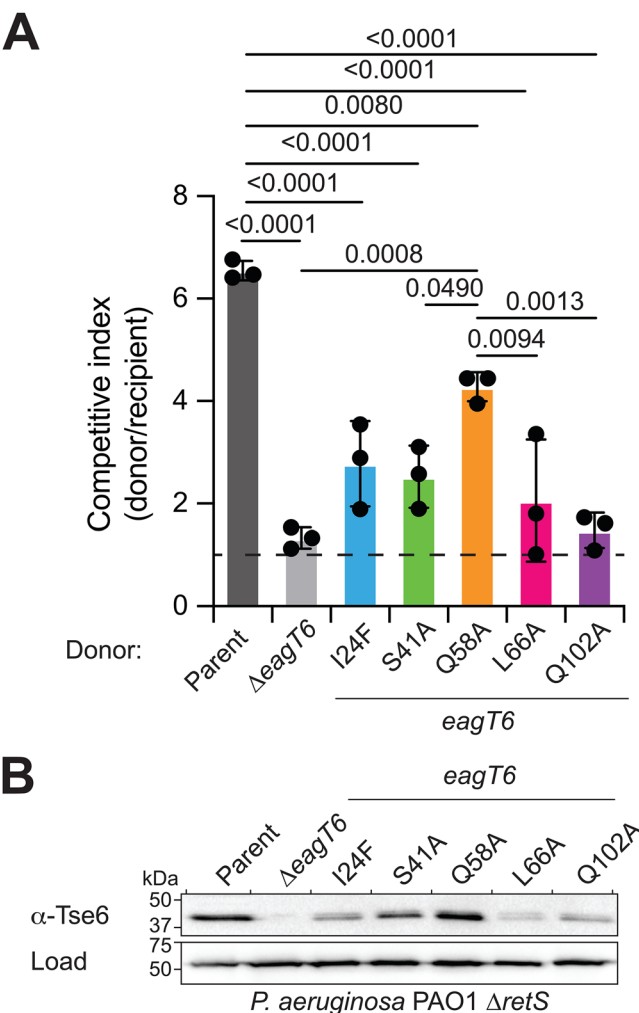

**Fig. 6 | Intraspecific competition experiments between *P. aeruginosa* PAO1 predator and prey strains. A** Bacterial competition experiment between strains harboring wild-type EagT6 and EagT6 point variants. The competitive index is calculated as the change (final/initial) in the donor to recipient ratio. The parent is PAO1 *ΔretS*. Horizontal dashed line at a competitive index of 1 indicates the line of no competitive advantage. Statistically significant differences are shown by *p*-value and error bars represent SDM of *n* = 3 biological replicates. All other pair-wise comparisons are not statistically significant (*p* > 0.05). **B** Western blot analysis of Tse6 expression in the indicated *P. aeruginosa* PAO1 strains. Parent is PAO1 *ΔretS*. RNA polymerase was used as a loading control.

### A crystal structure of SciW-TMD L66A suggests a TMD release intermediate

To further probe the molecular details of Eag-TMD complex stability, we attempted to crystallize a point variant Eag-TMD complex. We were able to obtain diffracting crystals of SciW-TMD L66A and obtain phases by molecular replacement (Fig. 7). Two SciW-TMD complexes were present in the asymmetric unit and all chains exhibited very high B-factors suggesting conformational dynamics (Supplementary Fig. 12A and Table 2). Specifically, one SciW-TMD L66A complex (chains D–F) showed several areas of missing density and the TMD was not confidently modelled. However, the other SciW-TMD L66A complex (chains A–C) showed strong density for the TMD (Supplementary Fig. 12B). Strikingly, the conformation of the bound TMD is significantly different from the wild-type complex (Figs. 7 and 3A). Given the high B-factors, lack of strong TMD density for one complex, and the observed alternate TMD conformation, the captured SciW-TMD L66A structure likely represents a TMD release intermediate.

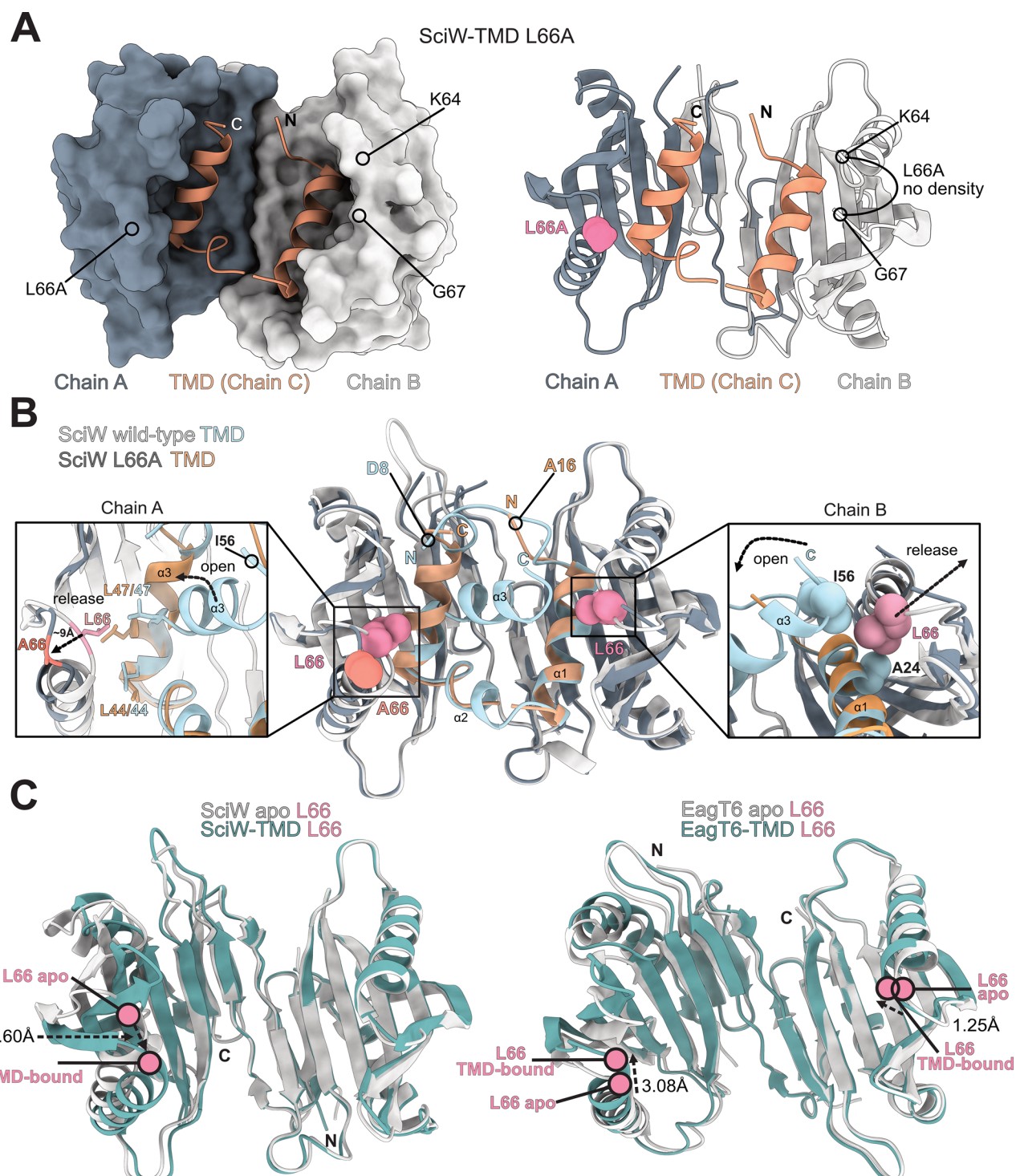

**Fig. 7 | An X-ray crystal structure of SciW-TMD L66A captures a TMD release intermediate. A** Structure of chains A−C. SciW is shown as a molecular surface and the TMD of Rhs1 in dark salmon (left). Cartoon ribbon diagram of chains A−C with the position of L66A indicated by space-filing in pink (right). The unmodelled loop region containing L66A in chain B is also indicated. **B** Structural comparison of wild-type SciW-TMD (grey-light blue) and SciW-TMD L66A (dark grey-dark salmon). Boxes show a zoom-in of the conformational changes due to the L66A variation at chain A (left) and chain B (right). Wild-type L66 is shown in dark red. L66A (A66) is shown in orange-red. Residues at the N-terminus of the SciW-TMD and SciW-TMD L66A complexes are indicated. **C** Structural overlay of Eag apo and TMD-bound chaperones SciW (left) and EagT6 (right). The change in position of L66 (pink-red) from apo Eag chaperone (white) to TMD-bound chaperone (teal) is indicated in angstroms (Å).

SciW-TMD L66A retains a dimeric structure similar to wild-type; however, the loop containing L66A at the tip of each claw is in a different conformation (Figs. 7A and 3). There is no observable electron density for the loop between SciW residues K64 and G67 of chain B

(Fig. 7A, right). Furthermore, the TMD is also no longer in a twisted asymmetric conformation. In wild-type SciW, the TMD (blue) adopts an asymmetric binding mode, making non-equivalent contacts with the chaperone dimer (Fig. 3, Supplementary Fig. 4, and Fig. 7B)[27].

**Table 2 | Data collection and refinement statistics for SciW-TMD L66A**

| Data collection | SciW-Rhs1(1–59) L66A |
|---|---|
| Wavelength (Å) | 1.1808 |
| Space group | C 1 2 1 |
| Cell dimensions | |
| a, b, c (Å) | 90.67 52.02 177.45 |
| α, β, γ (°) | 90 99.76 90 |
| Resolution (Å) | 45.16–3.05 (3.26–3.05) |
| Total reflections | 104432 (19002) |
| Unique reflections | 15847 (2884) |
| CC (1/2) | 0.996 (0.492) |
| $R_{merge}$ | 0.151 (1.948) |
| $R_{pim}$ | 0.064 (0.816) |
| I/σ | 8.2 (0.9) |
| Completeness (%) | 99.8 (100.00) |
| Redundancy | 6.6 (6.6) |
| **Refinement** | |
| $R_{work}$ / $R_{free}$ (%) | 26.1/29.8 |
| Average B-factors | 175.8 |
| Protein | 176.0 |
| Water | 143.5 |
| No. atoms | |
| Protein | 4877 |
| Water | 38 |
| Rms deviations | |
| Bond lengths (Å) | 0.002 |
| Bonds angles (°) | 0.63 |
| Ramachandran plot (%) | |
| Total favoured | 95.33 |
| Total allowed | 4.34 |
| PDB code | 9MVG |

Statistics for the highest resolution shell are shown in parenthesis.

Instead, the TMD folds as two symmetric helices packed similarly into the bottom of the L66A chaperone claw (Fig. 7A).

When comparing the SciW-TMD L66A structure to the wild-type complex, we observe that the L66A variant is unable to form critical knob-in-hole and hydrophobic packing interactions with the TMD (Fig. 7B). First, L66A in chain B no longer forms a knob-in-hole with Rhs1-TMD A24 in helix α1 or packs against residue I56 in helix α3 due to the shorter sidechain of the substituted alanine. This in turn frees SciW loop K64 to G67 and releases the C-terminus of the TMD from the chaperone, which causes the TMD to open outwards from its normal twisted conformation (Fig. 7B, chain B). Simultaneously in chain A, L66A no longer packs against TMD residues L44 or L47 in helix α3. The loop containing A66 moves ~9 Å away, also releasing the TMD to open in a hinge-like fashion, forming an α-helix that fills the chaperone binding pocket. This is in part due to TMD residue L47 moving into the space that the SciW L66 side chain previously occupied. Additionally, the bound N-terminal prePAAR region (residues 8–13) of the TMD is pushed out and released from SciW. We note that the prePAAR must be free to bind the PAAR for secretion[27]. A movie demonstrating the conformational change is provided (Supplementary Movie 1).

The captured SciW-TMD L66A dissociation intermediate likely represents a biologically relevant conformation, particularly when compared to wild-type SciW and EagT6 complexes. Figure 7C shows the conformational changes involving residue L66 that are required for effector TMD binding. In both Eags, the position of L66 is shifted

significantly inward to interact with the TMD. However, only in EagT6 does L66 in both chains appear to move. Given the importance of L66 for TMD binding and dissociation, the repositioning of L66 upon TMD binding may reflect a reversible conformational switch. Here, inward movement of L66 facilities TMD binding and outward movement of L66 disrupts the knob-in-hole packing interaction to promote TMD dissociation.

Our structural data also shows why L66 is important for SciW-TMD complex stability and in agreement with our φ-value analysis, suggests a partially ordered transition state during TMD dissociation. Given that residue L66 is critical for EagT6-TMD complex stability and that L66A allows the visualization of SciW-TMD unfolding intermediate, a conformational change to move L66 away from the TMD is likely central to the mechanism of TMD release for both classes of Eag chaperones.

## Discussion

Here, we use a combination of biophysical, molecular genetics, and structural biology methods to investigate the mechanism of effector transmembrane (TMD) binding to Eag chaperones. Our work shows that the Eag chaperones SciW and EagT6 exhibit an extremely strong binding affinity (tighter than antibody-antigens) for the TMDs of their cognate effectors Rhs1 and Tse6 (Figs. 1 and 2). As chaperones sequester TMDs by mimicking alpha-helical membrane packing, we designed point variants to disrupt these contacts and probe how the TMD might dissociate from an Eag (Fig. 3). Our quantitative thermodynamic and kinetic measurements demonstrate that the disruption of bifurcated hydrogen bonds (Q58) or knob-in-hole packing interactions (L66) between the Eag and TMD is sufficient to cause EagT6 to release the TMD both in vitro and in vivo (Figs. 4–6). Additionally, a co-crystal structure of SciW-Rhs1 L66A reveals an intermediate TMD-release state. The structure shows that the loss of key knob-in-hole packing interactions drastically changes the conformation of the TMD within the Eag and increases the dynamics of the complex (Fig. 7). Given that only small perturbations in Eag-TMD complexes drastically lowers the binding energy and leads to a large conformational change in the TMD, our work suggests that the mechanism of TMD release from an Eag only requires a subtle conformational change to disrupt one or two key Eag-TMD contacts.

Based on the collective findings from our biophysical and structural analyses, we propose a mechanistic model for TMD (effector) release (Fig. 8). Initially, the Eag chaperone binds the TMD of its cognate effector in the cytoplasm and facilitates PAAR domain binding to the tip of VgrG. This requires that the prePAAR motif binds and completes the effector PAAR domain[27], which likely includes prePAAR dissociation from the Eag (Fig. 8A). We note that for both SciW-TMD and Eag-TMD complexes, the prePAAR is dynamic and has been observed to adopt different conformations between the asymmetric units of our past crystal structures[27]. However, the fact that Eag-effector-VgrG complexes can be readily isolated and visualized by cryoEM from Gram-negative bacteria shows that PAAR and VgrG binding does not trigger Eag dissociation[27,31,37]. Furthermore, this strongly suggests that the dissociation catalysts is not cytoplasmic but specific to the T6SS apparatus. We therefore hypothesize that Eag chaperone removal occurs after the Eag-effector-VgrG complex is loaded into the baseplate and within the baseplate/membrane environment (Fig. 8B). Upon T6SS firing, the baseplate undergoes a conformational change[64]. This allows an apparatus component to contact the Eag-TMD complex directly, inducing the Eag chaperone to open and release the TMD (Fig. 8C, D). Our data indicates that the induced conformational change moves L66 and Q58 away from the TMD. Importantly, residue L66 is critical to Eag-TMD complex stability and undergoes a conformational change to initially bind the TMD[27]. Disruption of the critical knob-in-hole and packing interactions provided by L66 and Q58 lowers the incredibly high Eag-TMD binding energy

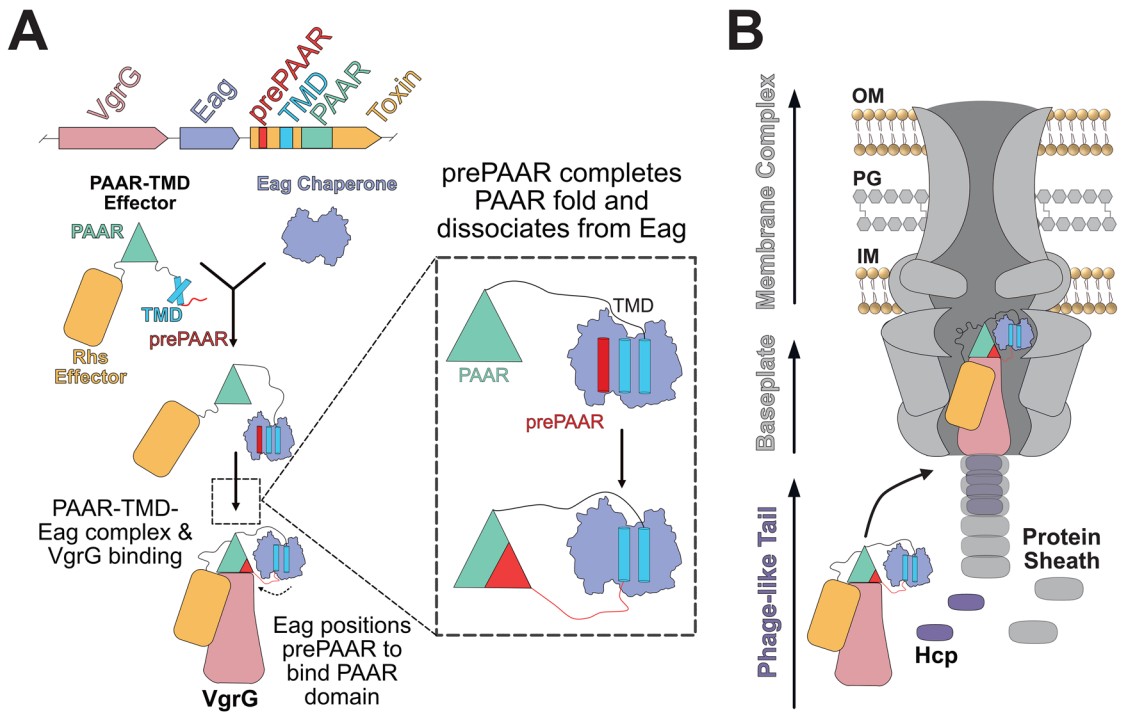

**1) Eag co-translated with PAAR-TMD effector and binds VgrG in cytoplasm**

**2) Eag-effector-VgrG complex is loaded into T6SS baseplate**

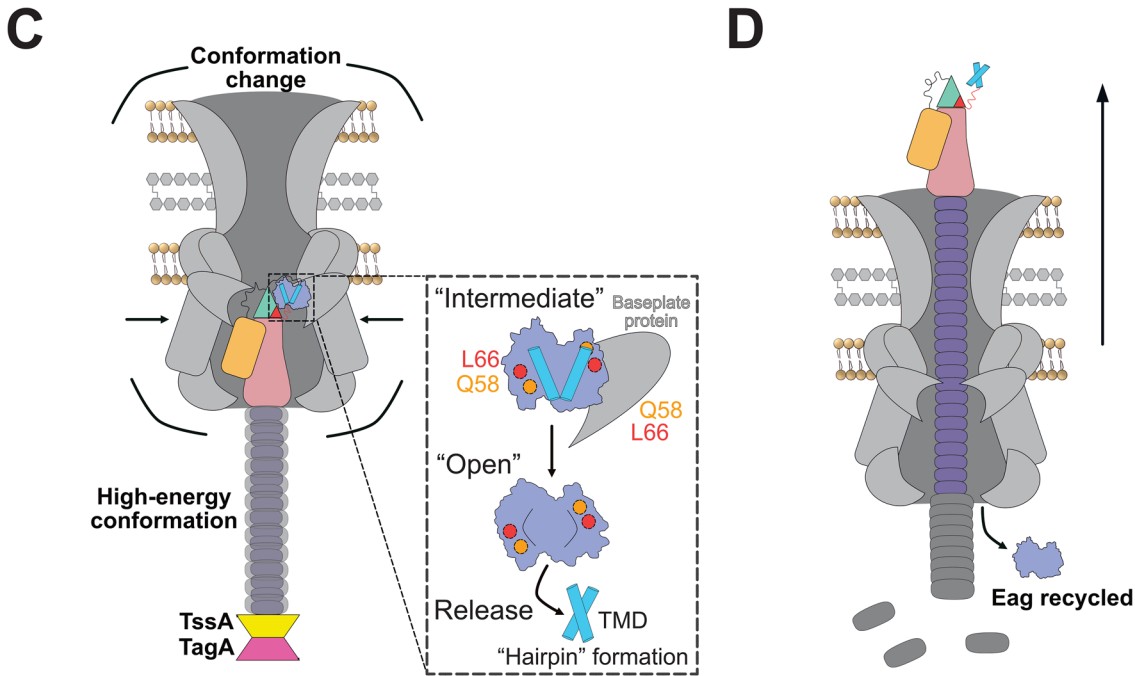

**3) Conformational change in baseplate contacts Eag, inducing a conformational change that disrupt L66 and Q58 interaction with TMD**

**4) Contraction of sheath and effector secretion**

**Fig. 8 | Proposed model of Eag chaperone disengagement and effector TMD release. A** Eag chaperones (purple) are co-translated with their multi-domain effectors (yellow), binding both the effectors TMDs (blue) and the N-terminal prePAAR region (red), thereby facilitating complex formation with VgrG in the cytoplasm. For proper folding and VgrG engagement, the prePAAR motif (red) must associate with the PAAR domain. The Eag helps position the prePAAR to bind the PAAR domain and associate with VgrG (dashed box). **B** The Eag-effector-VgrG complex is loaded on to the T6SS for secretion. **C** During T6SS firing, a conformational signal is transmitted from the membrane complex to the baseplate, which contacts the Eag-TMD complex and induces a conformational change intermediate in the Eag chaperone. This repositing disrupts the critical knob-in-hole packing interaction at L66 and bifurcated H-bond at Q58, moving the residues away from the TMD. Disruption of this interaction destabilizes the Eag-TMD interface causing the Eag to open, enabling effector TMD release (dashed box). **D** Upon T6SS sheath contraction, the unbound TMD effector is secreted and the Eag is recycled for a subsequent round of effector loading and secretion.

and increases the rate of TMD dissociation. This model also includes that the extremely large energy of T6SS firing[65] may be used to overcome the high binding energy of the Eag-TMD complex. However, SciW-TMD (class 1) and EagT6-TMD (class 2) proceed through different folding transition state intermediates, where SciW requires more than disruption of the L66 knob-in-hole.

Currently, it is unknown which T6SS protein might bind the Eag and catalyze TMD dissociation. Our model hypothesizes that a T6SS structural component of the baseplate or membrane complex, such as TssK, TssL, or TssM, binds the Eag chaperone[66,67]. This binding could then induce the conformational change in the Eag as described by our biophysical data and T6SS firing coupled model. Alternatively, TssM possesses ATPase activity, which phosphorylates TssL as part of Hcp recruitment[68] and could provide energy to dissociate the TMD. However, our biophysical data clearly demonstrates that the energetics of TMD release only requires outcompeting one Eag residue interaction (L66), at least for EagT6. Residue L66 is on a solvent accessible loop on the very end of the Eag helix that changes conformation to bind the TMD. Given this, a T6SS protein only needs to bind the Eag and outcompete L66 to lower the energy of TMD binding and allow dissociation. Therefore, we favor the T6SS firing coupled model (Fig. 8). Finally, the T6SS internal apparatus is shielded from the bulk solvent[66]. This could result in an environment that alters Eag sidechain pKas, disrupting bifurcated hydrogen bonds or outcompetes the van der Waal knob-in-hole interactions with the TMD. Regardless, additional experimentation is required to determine if a T6SS protein induces Eag chaperone removal from the effector.

Based on our data, the precise mechanistic details for TMD release differ between SciW-TMD and EagT6-TMD. In SciW, the TMD adopts a twisted conformation compared to an ordered anti-parallel helix found in EagT6[27]. This includes a helical conformation for the prePAAR bound to EagT6, while the prePAAR makes minimal contacts with SciW (Fig. 3 and Supplementary Fig. 4). These structural differences likely affect the mechanism of TMD dissociation. For EagT6-TMD, a conformational change to disrupt L66 is sufficient to displace the anti-parallel TMD helixes but only results in a release intermediate conformation for SciW. Dissociation of the TMD from SciW requires more than just disruption of L66. Residue Q58 is part of the α-helix that ends with the L66 loop, which also moves to accommodate TMD binding. This suggests that a conformational change that moves L66 would also displace Q58 (and other contacts) and therefore may allow TMD dissociation from either effector class. Alternatively, the conformation seen in the SciW L66A variant occurs first creating a short-lived intermediate, followed by disruption of other contacts. Additionally, residue Q106/2 is essential for the interaction of EagT6 with the TMD but contributes minimally to SciW-TMD complex stability. Residue Q106/2 is at the base of the Eag palm and does not alter conformation upon binding in either chaperone. However, for EagT6 Q102 makes contacts with both the prePAAR and the hydrophobic TMD helices but makes no prePAAR contacts in SciW. In line with this, the drastic binding energies between SciW and EagT6 may be in part due to interaction with the prePAAR. For EagT6, the prePAAR occupies one full binding face on a monomer, whereas for SciW, the hydrophobic TMD helices fill the chaperone. Moreover, the dynamic nature of the prePAAR suggests it is not tightly bound to the EagT6 chaperone[27]. Taken together, this may explain why disruption of L66 is sufficient to dissociate the TMD from EagT6, but additional energy is required to disrupt SciW-TMD.

For SciW-TMD, the higher complex stability and TMD intermediate observed by our biophysical data may act as an energetic guardrail. Such an energetic guardrail may also have biological relevance directly related to the structure of class 1 and class 2 prePAAR effectors[27]. EagT6 binds a class 2 effector that has 2 TMD regions. This requires 2 Eags to stabilize the TMDs and prevent their membrane-penetrative conformation. In contrast, SciW binds a class 1 effector that has only 1 TMD region, which would not require the extra energy barriers to remove a second Eag then complex with another TMD. Given this, it is likely that class 1 complexes have evolved higher stability than class 2 complexes to assure inhibition of TMD folding. It is important to note that we lack a crystal structure of EagT6-TMD2 and AlphaFold3 fails to predict a confident model (pLDDT <65 for TMD2). Our mechanistic dissociation data currently only applies to EagT6-TMD1. However, L66A is sufficient to inhibit secretion in vivo strongly supporting our TMD1 dissociation model and importance of TMD1 binding.

Our biophysical data also provides additional insight to understand Eag-TMD dissociation in the timeframe of T6SS assembly. The EagT6-TMD and SciW-TMD complexes have lifetimes on the order of ~18 s and ~15 h, respectively. Removal of the knob-in-hole and packing interactions of L66 reduces the TMD-bound lifetime to ~2 h for SciW and release of the TMD is unmeasurably fast for EagT6 (Fig. 5 and Table 1). Given that the T6SS can assemble and fire on the order of seconds to minutes[57–59] our biophysical data fits well for EagT6-TMD within the known time frame of T6SS kinetics. Additionally, our data supports that the Tse6 TMD release mechanism is two-state. In contrast, the release of the TMD from SciW is too slow to depend only on the removal of one knob-in-hole interaction. The SciW Q58 variant accelerates complex dissociation to ~16 min also supporting that release of the TMD from SciW requires a more substantive disruption than EagT6 as proposed in our biophysical model. Alternatively, this may also suggest a multi-step mechanism for TMD release from SciW. This hypothesis is supported by our SciW-TMD L66A crystal structure that captures a TMD release intermediate and the DSC data that shows a thermal unfolding intermediate for the TMD (Figs. 1C and 7). However, our chemical unfolding data for SciW is readily modelled by a 2-state mechanism, suggesting that further kinetic experiments are required to resolve if other SciW-TMD unfolding intermediates exist. Regardless, the data strongly supports that for both chaperones release of the TMD includes a conformational change to displace L66. Overall, our work has provided a detailed biophysical description of the energetics of Eag-TMD complex stability, Eag-TMD effector class specificity, and the mechanism for TMD release.

## Methods

### Molecular cloning of protein expression constructs

Wild-type Eag chaperones SciW from *Salmonella* Typhimurium (Eag: SL1344_0285) and EagT6 from *Pseudomonas aeruginosa* (Eag: PA0094) and their respective point variants were commercially synthesized from Twist Bioscience in the vector pET-29b(+). The Eag point variant SciW Q106 was commercially synthesized in multiple cloning site 2 of pETDuet-1 (NdeI/XhoI) (Genscript). A C-terminal VSV-G tag for detection by western blot was added using restriction sites NdeI and Xho1. Eag-TMD complexes SciW-Rhs1$_{NT}$ (Eag: SL1344_0285, Rhs1 (1–59)$_{TMD}$:SL1344_0286) and EagT6-Tse6$_{NT}$ (Eag: PA0094, Tse6 (1–61)$_{TMD}$: PA0093), were created previously in the vector pETDuet-1 in Ahmad et. al.[27]. C-terminal VSV-G-tagged Eag chaperones were both cloned into multiple cloning site 2 (NdeI/XhoI) and N-terminal domains of the effectors (TMDs) were cloned into multiple cloning site 1 with a C-terminal His$_6$-tagged (NcoI/HindIII). Eag-TMD point variants were created using Q5 Site-directed mutagenesis kit (New England Biolabs (NEB)) by amplifying the full-length construct using primer pairs outlined in Supplementary Table 4. Primers were generated from Integrated DNA Technologies (IDT) and constructs were confirmed by Sanger sequencing (SickKids Toronto). The complete set of strains and plasmids used in this study are outlined in Supplementary Tables 5 and 6.

### Protein expression and purification of Eag chaperones

Eag chaperones and their point variants were over expressed in *E. coli* BL21(DE3)-Gold cells using Lysogeny Broth (LB) culture medium and supplemented with 50 μg/mL kanamycin. Cells were grown at 37 °C in

a shaking incubator overnight and the next day diluted in 3 liters of LB broth and grown to OD600 of 0.6. Protein expression was induced by the addition of 1 mM isopropyl β-d-1-thiogalactopyranoside (IPTG) and cells were further incubated for 20 h overnight at 20 °C. Next, the cells were harvested by centrifugation at $4200 \times g$ for 30 min and resuspended in Eag lysis buffer (50 mM Tris pH 8). PMSF and MgCl$_2$ were added at final concentrations of 1 mM and 10 mM, respectively, along with a small amount of DNase. The cells were then lysed using an Emulsiflex-C3 High Pressure Homogenizer (Avestin) and centrifuged at $17,000 \times g$ for 25 min at 4 °C. The supernatant was passed over a Q-sepharose gravity column (GE Healthcare) equilibrated with $2 \times$ column volumes of Eag lysis buffer for purification by ion exchange. The column was washed with $3 \times$ column volumes of lysis buffer before being subjected to a stepwise gradient of $1 \times$ column volumes of Eag lysis buffer containing increasing concentrations of NaCl as follows: 50 mM NaCl, 100 mM NaCl, or 1 M NaCl. Fractions from the 50 mM elution step (50 mM Tris pH 8.0, 50 mM NaCl) were concentrated to 2 mL by centrifugation using a 3 kDa concentrator (Amnicon). The concentrated elution was further purified by size-exclusion chromatography (SEC) using a HiLoad (16/600) Superdex75 column (Cytvia) with an AKTApure (Cytvia). The SEC column was equilibrated with gel filtration buffer (50 mM Tris pH 8.0, 250 mM NaCl, 1 mM 2-Mercaptoethanol (BME)) before samples were injected. All SEC purifications were performed at 4 °C. To confirm purity, fractions were run on a 13% sodium dodecyl sulfate-polyacrylamide electrophoresis (SDS-PAGE) gel and visualized with Coomassie Brilliant Blue. Selected fractions were further concentrated and flash frozen in liquid nitrogen for further use.

### Protein expression and purification of Eag-TMD complexes
Eag-TMD complexes and their point variants were expressed in BL21(DE3)-Gold cells using LB culture medium supplemented with 100 μg/mL ampicillin. Eag-TMD complexes and point variants were lysed and harvested as described above. For Eag-TMD complexes, the Eag-TMD lysis buffer consisted of 50 mM Tris pH 8.0, 500 mM NaCl, and 20 mM imidazole. Protein complexes were purified by affinity chromatography using a nickel-NTA agarose gravity column (Goldbio) equilibrated with $1 \times$ column volume of lysis buffer. The bound proteins were washed with $3 \times$ column volumes of lysis buffer before being eluted by $1 \times$ column volume of elution buffer (50 mM Tris pH 8, 500 mM NaCl, 500 mM Imidazole). Following metal-affinity chromatography, Eag-TMD complexes and point variants were further concentrated and purified by SEC using the same protocol as the Eag chaperones. For all Eag-TMD complexes and point variants, the gel filtration buffer was 50 mM Tris pH 8.0, 250 mM NaCl, 1 mM BME. Finally, pure fractions as assayed by Coomassie-stained SDS-PAGE gel were concentrated and flash frozen in liquid nitrogen for further crystallization and biophysical experiments.

### Dynamic light scattering (DLS)
DLS experiments were performed with a Prometheus Panta (Nano-Temper) in DLS buffer (50 mM Tris pH 8.0, 250 mM NaCl, 1 mM BME) at 25 °C and a protein concentration of 0.5 mg/ml. Each measurement lasted for 15 s and was repeated 10 times in triplicate thus providing accumulation of 30 measurements. The data were analyzed with the PR.Panta Analysis software version 1.10.2 (Nanotemper) using the intensity distributions for all subsequent evaluations.

### Nano differential scanning fluorimetry (nanoDSF)
Thermal unfolding experiments of purified wild-type and variant Eag chaperones and Eag-TMD complexes were performed in a Prometheus NT.48 instrument (NanoTemper Technologies Inc.). Protein samples were diluted in gel filtration buffer (50 mM Tris pH 8.0, 250 mM NaCl, 1 mM BME) to 0.5 mg/mL and loaded in 10 μL nanoDSF Grade Standard capillaries in triplicate. Samples were heated from 20 °C to 95 °C with a

temperature ramp of 1.0 °C/min. Fluorescence intensity was monitored at 330 nm and 350 nm after excitation of tryptophan at 280 nm. Melting temperatures ($T_m$) were calculated from the first derivative of the 350/330 nm emission ratio. Data was analyzed with the PR. ThermControl software (NanoTemper Technologies Inc.) and Graph-Pad Prism version 10.0.2, GraphPad Software, Boston, MA, USA.

### Differential scanning calorimetry (DSC)
Purified wild-type Eag chaperones and Eag-TMD complexes were dialyzed overnight into buffer containing 50 mM PBS pH 7.5, 250 mM NaCl, and 0.1 mM Tris(2-carboxyethyl)phosphine hydrochloride. For DSC measurements, an equal volume of 0.5 mg/mL protein solution in DSC buffer as well as a reference solution containing DSC buffer without the addition of protein, were prepared. Both the protein and reference solutions were degassed for 10 min under vacuum to ensure removal of air bubbles prior to loading of DSC. Experiments were performed using a Nano DSC (TA Instruments, New Castle, DE) with a 0.3 mL capillary cell volume. Samples were equilibrated at 20 °C for 600 s and heated to 95 °C with a scan rate of 1 °C per minute at a constant pressure of 3 atm. A background scan was performed by loading both sample and reference cells with reference solution prior to the sample scan. Data was analyzed using NanoAnalyze (TA Instruments, New Castle, DE). The sample background was subtracted, and a sigmoidal baseline was applied to the data. The data was fit to a two-state scaled model for the transition.

### Steady-state tryptophan fluorescence spectral analysis
For chemical denaturation experiments using the Prometheus NT.48 (fixed 330 and 350 nm), purified proteins were denatured in increasing concentrations of either urea of guanidine hydrochloride (0.08 M to 7.6 M urea, Δ dilution = 0.3 per capillary; 0.6 M to 5.6 M guanidine hydrochloride, Δ dilution = 0.24 per capillary). For each experiment, capillaries containing 24 different concentrations of denaturant and 0.5 mg/mL of protein were loaded in triplicate into a Prometheus NT.48 instrument (NanoTemper Technologies Inc.). Data was analyzed with the PR. ChemControl software (NanoTemper Technologies Inc.) and GraphPad Prism version 10.0.2 GraphPad Software, Boston, MA, USA.

For quantitative spectral analysis of wild-type Eags and both Q58A and L66A variants, intrinsic protein fluorescence was measured in the presence of increasing concentrations of guanidine hydrochloride (GdmCl) (1.50 M, 2.0 M, 3.0 M, 3.5 M, 4.0 M, 4.5 M, 5.0 M, 5.5 M, 6.0 M). Concentrated protein was dialyzed in 1 L of buffer (25 mM Tris pH 7.0) for 1.5 h to ensure adequate buffer exchange. The protein was then diluted to the appropriate stock concentration. Steady-state fluorescence spectra were measured on a Fluorolog-3 Horiba Jobin Yvon spectrofluorometer (Edison, NJ) using a $10 \times 2$ mm$^2$ quartz cuvette to hold the sample. The samples were excited at 280 nm, the excitation and emission slits were set to a 2 nm bandpass. A series of 500 μL samples with a final concentration of 1 μM protein dimer (2 μM monomer), were prepared at various GdmCl concentrations. All samples were prepared in triplicate. The samples were equilibrated at room temperature for 1 h prior to the scan. The final denaturant concentrations were determined by refractive index using C10 Abbe Refractometer (VEE GEE Scientific) post data acquisition. The data were analyzed with Sigma Plot version 12.3 (2013) Systat Software Inc, (San Jose, CA) software and OriginPro Version 2025 (OriginLab Corporation, Northampton, MA, USA).

### Calculation of free energy of Eag unfolding and TMD binding from steady-state tryptophan fluorescence
If it is assumed that the chaperones SciW and EagT6 unfold through a two-state mechanism with an equilibrium constant of $K_{unfolding}$:

$$chaperone_{dimer,folded} \rightleftharpoons 2\ monomer_{unfolded} \qquad (1)$$

The fraction of tryptophan residues in the unfolded state "α" can be related to the measured $\lambda_{max}$ through the equation[47]:

$$\alpha = \frac{\lambda_{max} - \lambda_f}{\lambda_u - \lambda_f} \qquad (2)$$

The denaturant concentration dependencies of $\lambda_f$ and $\lambda_u$ are given in Supplementary Fig. 3. Allowing us to define the equilibrium constant:

$$K_{unfolding} = \frac{2c_0\alpha^2}{(1-\alpha)} \qquad (3)$$

Where $c_0$ is the total concentration of Eag monomer in solution, $\lambda_f$ is the emission maximum wavelength of the folded protein and $\lambda_u$ is maximum wavelength of the unfolded protein.

Next, $K_{unfolding}$ is related to the change in free energy by:

$$\Delta G_{unfolding} = -RT \ln K_{unfolding} \qquad (4)$$

Which allows the determination of the standard change in unfolding free energy in buffer $\Delta G^o_{unfolding}$, assuming a linear free energy dependence upon denaturation concentration[44]:

$$\Delta G_{unfolding} = \Delta G^o_{unfolding} + m[GdCl] \qquad (5)$$

Based upon this, the denaturation concentration dependence of α (fraction unfolding) is given by:

$$\alpha = \frac{-K_{unfolding} + \sqrt{K^2_{unfolding} + 8c_0 K_{unfolding}}}{4c_0} \qquad (6)$$

In which $K_{unfolding} = e^{-\frac{\Delta G^o_{unfolding} + m[GdCl]}{RT}}$.

If the TMD is assumed to selectively bind to the folded form of the Eag with an association constant $K_{TMD}$, the effect of TMD binding upon the standard unfolding free energy for the chaperone can be estimated using the relationship[49,50]:

$$\Delta G^o_{unfolding, TMD\ bound} = \Delta G^o_{unfolding, apo} + RT \ln(1 + K_{TMD}[TMD]) \qquad (7)$$

Where $K_{TMD} = \frac{[chaperone - TMD]}{[chaperone][TMD]}$, if the TMD binds the chaperone tightly, the majority of the chaperone is bound to the TMD and $[chaperone - TMD] \approx 0.5\,c_0$, $[TMD] = \sqrt{\frac{0.5c_o}{K_{TMDol}}}$ and $K_{TMD}[TMD]$ is significantly larger than one. We can then simplify Eq. (7) to:

$$\Delta G_{unfolding, TMD\ bound} \approx \Delta G_{unfolding, apo} + RT \ln(\sqrt{0.5c_0 \times K_{TMD}}) \qquad (7a)$$

Rearranging this equation we obtain:

$$\Delta G^o_{unfolding, TMD\ bound} - \Delta G^o_{unfolding, apo} \approx 0.5RT\ln 0.5c_0 + 0.5RT\ln K_{TMD}$$
$$= 0.5RT\ln 0.5c_0 - 0.5\Delta G^o_{binding} \qquad (7b)$$

## Stopped-flow kinetics

Unfolding kinetics were performed using Applied Photophysics SX-20 (Surrey, UK) stopped-flow fluorescence instrument (dead time <2 ms). To ensure the appropriate dilution of denaturant, asymmetric mixing was set up using 2.5 mL and 250 µL drive syringes purchased from Delta photonics (Ottawa, CA). Purified wild-type and variant Eag chaperones and complexes were first diluted to a concentration of 10 µM in gel filtration buffer (50 mM Tris pH 8.0, 250 mM NaCl, and 1 mM BME). Purified protein was diluted in a 1/10, v/v, mixing ratio with guanidine hydrochloride (GdmCl) in 50 mM Tris, 250 mM NaCl, 1 mM

BME of the desired concentration. The excitation wavelength was set to 280 nm, and the emission was monitored using a 330 nm Bandpass filter (FWHM 10 nm). All data was collected at 20 °C. At least 12 unfolding traces were collected at each desired concentration of GdmCl and averaged. The final denaturant concentrations were determined by refractive index using C10 Abbe Refractometer (VEE GEE Scientific) post data acquisition. All data was analyzed using Sigma Plot 12.3 (2013) Systat software (San Jose, CA).

## Calculation of unfolding rate constants and psi-value analysis from stopped-flow kinetics

The EagT6 and SciW relaxation profiles can be described as a mono-exponential decay:

$$F = F_0 e^{-k_{obs}t} \qquad (8)$$

Where $F$ is the measured fluorescence, $F_0$ is the fluorescence at time zero and $k_{obs}$ is the measured rate constant; at high denaturant concentrations $k_{obs} \approx k_{unfolding}$, $k_{unfolding}$ being the apparent unfolding rate constant. Using a chevron-type analysis, it can be assumed:

$$\ln k_{unfolding} = \ln k^o_{unfolding} + m'[denaturant] \qquad (9)$$

Where $k_{unfolding}$ is the unfolding rate constant measured at a given concentration of denaturant, $\ln k^o_{unfolding}$ is the unfolding rate constant in pure buffer, and $m'$ is the "kinetic m-value". We have measured time-dependent unfolding traces of all Eag proteins (WT and variants), in the apo and TMD-bound states and have plotted $\ln(k_{obs} \approx k_{unfolding})$ as a function of denaturant concentration in Fig. 5; analyzing this date via Eq. (7) yields the kinetic parameters $\ln k^o_{unfolding}$ and $m'$ as shown in Table 1.

For "φ-value analysis", if the free energy of the unfolding pathway's rate-limiting transition state is represented by $\Delta G^{\ddagger}_{unfolding} = -RT \ln k^o_{unfolding} + RT\ln A$ in which $A$ is the pre-exponential factor defined by transition statec theory and the φ-value for each mutant is defined as:

$$\varphi = \frac{\left(\Delta G^{\ddagger}_{unfolding}\right)_{WT} - \left(\Delta G^{\ddagger}_{unfolding}\right)_{mutant}}{\left(\Delta G^0_{unfolding}\right)_{WT} - \left(\Delta G^0_{unfolding}\right)_{mutant}} \qquad (10)$$

## P. aeruginosa strains and growth conditions

P. aeruginosa strains were derived from the sequenced strain PAO1. Cultures were grown in LB medium (10 g/L tryptone, 5 g/L yeast extract, 10 g/L NaCl), shaking at 37 °C. Solid media containing 1.5% or 3% (w/v) agar, as indicated. E. coli strains XL1 blue and SM10 were used for plasmid maintenance and conjugative transfer, respectively. E. coli strains were grown in LB at 37 °C in a shaking incubator. Cultures were supplemented with 15 mg/mL gentamicin (E. coli), 30 mg/mL gentamicin (P. aeruginosa), and 25 mg/mL irgasan (P. aeruginosa).

## DNA manipulation, plasmid construction, and strain generation for bacterial competition and secretion assays

All primers were synthesized by IDT. Phusion polymerase, restriction enzymes, and T4 DNA ligase were obtained from NEB. Sanger sequencing was performed by the Centre for Applied Genomics, The Hospital for Sick Children in Toronto, Ontario. Chromosomal mutations in P. aeruginosa were generated by double allelic exchange as previously described[69]. Approximately 500 bp flanks upstream and downstream of the region to be mutated were amplified by PCR and spliced together by overlap-extension PCR. Primers were designed such that the desired point mutation was contained within the overlap. The resulting amplicon was ligated into the pEXG2 allelic exchange vector, transformed into E. coli SM10, and introduced to P. aeruginosa

by conjugative transfer. Merodiploids were selected on LB agar containing 30 mg/mL gentamicin and 25 mg/mL irgasan and streaked on LB agar lacking NaCl containing 5% (w/v) sucrose for *sacB* counterselection. Strains that grow on sucrose but are gentamicin sensitive were screened by Sanger sequencing the PCR amplicon of the region of interest to confirm the presence of point mutations. A complete list of strains used in this study is outlined in Supplementary Table 6. Primers for the creation of Pseudomonas mutant strains are shown in Supplementary Table 7.

### Bacterial competition assays in *P. aeruginosa*

Assays were performed in co-culture as previously described[70]. Recipient strains harboured a constitutively expressed lacZ gene at a neutral site in the chromosome to distinguish them from donor strains upon plating on LB containing 40 mg/mL 5-bromo-4-chloro-3-indolyl-β-D-galactopyranoside (X-gal). Overnight cultures of indicated donor and recipient strains were diluted to an OD600 of 1.0 and combined in an initial donor/recipient ratio of 5:1(v/v), and the initial mixture was serially diluted to enumerate colony-forming units (CFU) by plating on media containing X-gal. $10\,\mu L$ of this mixture was spotted onto a nitrocellulose membrane on LB containing 3% (w/v) agar and cultured at 37 °C for 20 h. Co-cultures were resuspended in 1 mL LB and serially diluted to enumerate CFU. Data are presented as the fold change in the ratio of predator to prey at the end of the experiment relative to the beginning. Statistical significance was calculated by a one-way ANOVA on the log-transformed competitive indices using Tukey's correction for multiple comparisons between each mean. Western blot was done using a custom polycolonal antibody against Tse6[31] (1:3000 dilution) and a commercial anti-RNAP antibody (BioLegend)(1:5000 dilution).

### Bacterial competition assays in *S.* Typhimurium

Briefly, cultures of *S.* Typhimurium SL1344 (streptomycin-resistant) and *E. coli* DH5α carrying pET29b (kanamycin-resistant) were grown overnight in LB broth supplemented with the appropriate antibiotic. The following day, overnight cultures were washed three times with PBS and adjusted to an $OD_{600}$ of 0.4. The strains were then mixed at a 1:1 (vol/vol) ratio of wild-type or mutant *S.* Typhimurium to competitor *E. coli* DH5α. To determine the initial CFU ratio, the input mixture was serially diluted and plated on LB agar containing either kanamycin or streptomycin. Interbacterial competition assays were performed by spotting 10 μL of the mixed cultures onto 0.22 μm nitrocellulose on LB agar plates overlaid with either 0.05% ox-bile (Sigma) or PBS (control). After 48 h, cells were recovered from the membranes with an inoculating loop, resuspended in PBS, serially diluted, and plated on selective LB agar to enumerate final CFU counts. The competitive index was calculated as the fold change in the ratio of *S.* Typhimurium to *E. coli* between the initial and final CFU counts. Statistical significance was assessed using an unpaired *t*-test. A complete list of strains used in this study is outlined in Supplementary Table 4. The wild-type SL1344, Δ*sciW*, and Δ*clpV* mutant strains were a generous gift from Brian Coombes (McMaster University)[63].

### Protein crystallization

Purified SciW-Rhs1(1–59) L66A was screened for crystallization utilizing commercially available screens (Nextal) and a crystal Gryphon robot (Art Robbins Instruments). Crystals of SciW-Rhs1 (1–59) L66A grew in gel filtration buffer and 0.1 M MES pH 6, 0.2 M $MgCl_2$, and 20% (v/v) PEG 6000 20 mg/mL protein in a 1:1 mixture at 4 °C. Crystals were further optimized by micro-seeding using a SeedBead kit (Hampton Research).

### Data collection and refinement

A data set for SciW-Rhs1(1–59) L66A was collected at the Canadian Light Source beamline CMCF-BM (08B1). Protein crystals were first cryo-protected in 30% PEG 6000 before flash freezing directly in liquid nitrogen. Data was processed using XDS[71] and CCP4[72]. Initial phases were obtained from Phenix[73] using SciW-Rhs1(1–59) structure (PDBid 6XRR) as a model for molecular replacement. The structure was further built using Coot[74] and refined using Phenix, and TLS refinement[75]. Molecular graphics were generated using ChimeraX and GraphPad Prism version 10.0.2.

### Reporting summary

Further information on research design is available in the Nature Portfolio Reporting Summary linked to this article.

## Data availability

The X-ray structure and diffraction data reported in this paper have been deposited in the Protein Data Bank under the accession codes 9MVG. The previously published coordinates for SciW, SciW-Rhs1(1–59), EagT6, and EagT6-Tse6(1–61) are available at the PDB under the accession codes 6XRB, 6XRR, 1TU1 and 6XRF, respectively. Source data are provided with this paper.

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

## Acknowledgements
We would like to thank the laboratory of Jörg Stetefeld at the University of Manitoba for DLS instrument access. We also thank Brian Coombes (McMaster University) for providing *Salmonella* Typhimurium T6SS mutant strains. We would also like to thank beamline CMCF-BM at the Canadian Light Source, which is supported by the Canada Foundation for Innovation (CFI), the Natural Sciences and Engineering Research Council (NSERC), the National Research Council (NRC), the Canadian Institutes of Health Research (CIHR), the Government of Saskatchewan. This work was supported by a Canadian Institutes of Health Research (CIHR) project grant PJT-180450 to G.P., a Research Manitoba New Investigator Operating grant to G.P., and a Canadian Foundation for Innovation (CFI) grant 37841 to G.P. This work is also supported by Canadian Institute of Health Research (CIHR) grant PJT-175011 to J.C.W. and a National Sciences and Engineering Research Council of Canada (NSERC) Discovery Grant RGPIN-2024-06410 to M.K. M.V.S. is supported through a Research Manitoba Graduate Studentship and University of Manitoba Graduate Fellowship (UMGF). I.A. is also supported through a University of Manitoba Graduate Fellowship (UMGF). J.C., A.G., and K.S. are supported by PhD Canada Graduate Scholarships from NSERC.

## Author contributions
M.V.S. and G.P. conceived the study. M.V.S expressed, purified, and crystalized proteins. A.G. expressed and purified proteins. M.V.S. and I.A. performed biophysical experiments, and M.V.S., I.A., and M.K. analyzed all biophysical experiments. M.V.S. and G.P. solved and analyzed the crystal structure. J.C. performed bacterial competition assays and S.A. and J.C. constructed the *Pseudomonas* strains used in this study. K.S. performed bacterial competition assays with *Salmonella* strains. M.V.S., I.A., M.K., J.C.W., and G.P. wrote the paper. All authors provided feedback on the manuscript.

## Competing interests
The authors declare no competing interests.
