## [Transparent Peer Review file · Nature Communications]

Biophysical characterization of Eag chaperones suggests the mechanism of effector transmembrane domain release

Corresponding Author: Dr Gerd Prehna

Version 0:

Reviewer comments:

Reviewer #1

(Remarks to the Author)

The bacterial type VI secretion system (T6SS) is a potent antimicrobial weapon used by Gram-negative species. The T6SS can deliver numerous toxins with a broad range of biochemical activities, including enzymatic such as peptidoglycan hydrolases or lipases, as well as membrane pore forming. Targeting and loading of effectors onto the T6SS machine can involve a range of chaperones of which the Eag family has been involved in protecting hydrophobic and transmembrane domain (TMD) regions in toxins, such as nicely described between EagT6 and the *Pseudomonas aeruginosa* toxin Tse6. Here the authors studied in very fine detailed the molecular interactions between Eag and two T6SS effectors, T6SS and SciW from *Salmonella Typhimurium*. They performed in depth thermodynamic and biophysical characterization to monitor the high stability of the Eag-TMD interaction that prevents transmembrane domains to prematurely insert into membranes as well as the subsequent effective release of this interaction. To support their data, they investigate structural features of domain interactions and identify key residues essential for the stability of the complex and which are found in a knob-in-hole packing arrangement features. This is further validated by genetic studies in vivo demonstrating that lack of stability in the Eag-TMD by substituting these key residues results in absence of killing during interbacterial competition. Overall, the study is very technical and methodological, but very convincing although the gain in knowledge is rather incremental

Specific comments

The difference in lifetime and stability between the complex EagT6-Tse6 and Eag-SciW is astonishing, 18s vs 15h, respectively. Although the authors put these two complexes in different classes, it seems that the key residues involved in stability are the same in both complexes. Could this mean that other domains in the T6SS effector may still strongly influence the stability of the complex and possibly that additional partners (line 486) strengthen and weaken the interaction. For example, if Eag-SciW1 separation requires disruption of more than one contact (lines 369-370) then this may be catalyzed by additional partner. Do the Eags have other partners that could be identified for example through in vivo pull-down. Are there mutations that could be identified that would dramatically increase the lifetime of Eag-Tse6 or dramatically decrease lifetime of Eag-SciW1.

The authors suggest that the contact disruption within the Eag complexes occur by conformational change once the effector is in the T6SS baseplate (line 472). The firing and contraction mechanism associated with T6SS dynamic, involves a drastic conformation change in the baseplate from dome to starshaped. Would this contraction event contributing to the energy requires to disrupt Eag complexes. Could the Eag complex loaded in the T6SS nanomachine be modelled in the context of an extended or contracted version of the T6SS?

In figure 6 the most drastic impact for interbacterial competition is the Q102A substitution which is at least as impactful than the L66A substitution. The equivalent of Q102A substitution in the Eag-SciW1 complex was unsuccessful (lines 231-232). As such there is very little data or discussion on that particular residue in the main text. Yet could it be a difference that may contribute explaining the lifetime differences between Eag-Tse6 and Eag-SciW1. This may be worth to investigate and discuss further.

Reviewer #2

(Remarks to the Author)

Since my experience with the various biophysical techniques used in this manuscript is limited, I will only comment on the phenotypes related to T6SS function and impact of this paper on the field. This is overall a well written manuscript that addresses an important aspect of T6SS effector loading. The insights are interesting and such a detailed analysis is important for the field. I see no missing controls for the experiments testing phenotypes of the mutations generated based on the biochemical analysis. I suggest to add a figure with a cartoon like model for effector loading on the T6SS spike and removal of the chaperon (synthesis of the presented data and previous knowledge). This would help readers to imagine the whole sequence of events during T6SS assembly and effector loading.

Reviewer #3

(Remarks to the Author)

Effectors of the type VI secretion system (T6SS) are usually delivered with the ejected needle (Hcp) or its tip (VgrG/PAAR), either as direct fusions to these proteins or bound to them. Some effectors require chaperone proteins that can take various roles, but are usually required for delivery of their cognate effector, without being exported themselves. One such class of chaperones are the Eag chaperones, which in particular bind transmembrane domain (TMD) effectors. The authors of this manuscript had investigated the binding of Eag chaperones to a conserved motif found in TMD-containing effectors in eLife in 2020. In this manuscript, they follow up on this publication by addressing the obvious question how the Eag chaperones are removed from the effectors prior or during secretion. To this aim, the authors measure the stabilization of two Eag chaperones by the presence of the TMD of their respective effectors. They complement this analysis by structural analysis and a set of competition and expression experiments on a set of mutants.

The manuscript investigates an interesting question and the results appear solid and are well-presented (excluding the points raised in the minor comments). However, the reasoning and logical connections are hard to follow in various places, and it is unclear if the experiments (largely unfolding by chaotropic agents) allow for the conclusions on the T6SS assembly or export process. It is, therefore, difficult to assess whether the authors' model and conclusions are justified by the data.

Major comments:

1. In particular, it remains unclear at which time point the authors think the Eag chaperones are removed from the TMH effectors. The scheme in the authors' 2020 eLife paper shows the Eag chaperones as parts of the assembled T6SS prior to firing. This assumption is also retained throughout the manuscript, until, suddenly, in the description of the model (lines 463 onwards), Eag-TMD release is hypothesized to occur upon binding to the baseplate (which is not analyzed or even mentioned once in the results). The two scenarios are vastly different, as removal upon firing would likely imply mechanical stress, which is quite different from the scenario to which the measurements likely apply (GdmCl-induced unfolding).
2. The authors state that they have been unable to express TMD effectors without their chaperones and understandably, therefore, choose indirect methods to analyze the binding and unbinding. However, it is not immediately obvious that these methods allow to draw the conclusions made in the manuscript. To the best of my knowledge, changes in the Tm provide qualitative, but not quantitative binding affinities. Perhaps more importantly, this applies to equilibrium kinetics, but not to mechanical stress, as possible relevant for the removal of the T6SS chaperones. If the methods do allow for these conclusions, the underlying reasoning needs to be made clearer to the reader.
3. Negative controls, such as the addition of generic TM or of the respective non-cognate TMD, are missing for the key experiments.

Minor comments:

1. Why does Rhs1(1-59) aggregate on SDS-PAGE gels (Fig. S1 and S5), but not in the DLS measurements (Fig. S5)? Tse6(1-61) behaves different in this respect. Additional controls, beyond the fact that Rhs1(1-59) still binds to SciW, would provide evidence for its native binding, and the respective conclusions.
2. Is there a difference between chaperones and adaptors? If so, it should be explained, if not, this should be mentioned when mentioning adaptors (e.g. Tap adaptors).
3. The reason to focus on the first TMD of Tse6 (or a reference showing binding of this TMD only to EagT6) should be provided.
4. Lines 66/67: This statement does not apply to all effectors, and is therefore misleading in its current state.
5. The role of chaperones should be explained more clearly – lines 67-69 are rather vague and unspecific.
6. Line 123: Rhs should be called Rhs1, as elsewhere in the manuscript.
7. The dimer formation of Eag chaperones, one of their defining features, is not taken into consideration in the denaturation experiments (Fig. 1, 2), although likely to influence the results. Likewise, it is not taken into account in the equilibrium calculations described in lines 692-698. The authors seem to assume an essentially complete dimerization of the chaperones, but no evidence for this is provided.
8. How exactly does the difference in denaturation by GdmCl and Urea allow for the determination of the free energy of Eag-TMD unfolding (statement in lines 163-164)? Is urea used afterwards and what for?
9. Are lines 212-218 new results (as implied by the inclusion in the results part and lack of reference)? If they are instead a recapitulation of the results in ref. 27, they should be moved from the results part to the introduction, or this fact should be at least made clear.
10. The cellular concentration of Tse6 (which the authors, conceivably, assume to be determined by the binding to its chaperone) does not correlate well with the competitive index. For example, the Q58A mutant, in which the level is similar to the WT, has a strongly decreased fitness compared to the WT, and S41A, which retains expression, is doing worse than I24F, where almost no Tse6 could be detected. Can the authors explain these discrepancies?
11. What is meant by "the Eag chaperone catalyzes completion of the PAAR domain with the prePAAR motif" (lines 463-465, in the description of the model)? The expression is not contained in the listed ref. 27, and the mechanics described by it remain unclear.
12. There is a certain inconsistency and confusion in the gene identifiers given by the authors (lines 547-552 and

elsewhere):

- SL1344_0268, assigned to Salmonella SciW in line 547, encodes instead for TssC, the large subunit of the T6SS contractile sheath
- SL1344_0286, assigned to SctW elsewhere, seems to encode for an RHS protein instead
- SL1344_0285, assigned to Rhs1, encodes for a much smaller protein
- Likewise, the two protein identifiers may have been mixed up for the Pseudomonas genes

Reviewer #4

(Remarks to the Author)

Reviewer #5

(Remarks to the Author)

The manuscript by Van Schepdael et al. explores the mechanism of effector transmembrane domain (TMD) release from Eag chaperones, crucial for Type VI secretion system (T6SS) function. The authors use a combination of biophysical techniques and bacterial assays to demonstrate that small perturbations in Eag-TMD interactions, such as disrupting hydrogen bonds or knob-in-hole packing, can trigger TMD release. Their work underscores the essential role of conformational changes in Eag chaperones for the dissociation of TMDs, a critical step for effector secretion and bacterial fitness. Additionally, the X-ray co-crystal structure of the Eag-TMD complex captures an intermediate state of TMD release, offering a novel perspective on this process. This study contributes significantly to our understanding of how T6SS chaperones regulate effector folding and secretion. The insights provided in this study are important and should be of broad interest. While, to strengthen the work, the authors may need to address my concerns listed below.

Major concerns:

1. The study investigates the mechanism of Eag chaperone dissociation from effector proteins, but the extent to which the in vitro-derived dissociation mechanism reflects the actual dissociation process in bacterial cells is questionable. For example, the manuscript primarily examines the impact of different mutations on bacterial secretion but does not confirm whether these mutations specifically affect Eag dissociation, which then influences bacterial secretion and survival. The authors should include in vitro dissociation kinetics experiments and in vivo/in-cell binding assays to demonstrate the role of chaperones in regulating effector protein folding and their involvement in T6SS under physiological conditions.
2. The SciW-TMD L66A crystal structure presented in the study captures an intermediate state of Eag chaperone dissociation from TMD and supports the proposed dissociation model. However, this structure was obtained by introducing the L66A mutation to stabilize the dissociation intermediate. It is unclear whether this structure fully reflects the dissociation process under physiological conditions in bacterial T6SS, as wild-type chaperones in cells may dissociate differently. Could the structure of the wild-type chaperone dissociating from TMD under natural conditions be obtained by optimizing parameters such as temperature?
3. The study attributes the dissociation pathways of Class 1 (SciW-TMD) and Class 2 (EagT6-TMD) effector proteins to “the different roles of Q58/L66 contacts” and “structural differences in transition states.” However, the TMD sequences, overall conformations, and coupling with VgrG/PAAR differ significantly between the two effector classes, suggesting that other factors may regulate chaperone dissociation. The authors should include further experiments to confirm that the chaperone-TMD interactions are responsible, rather than other factors playing a key role in regulating T6SS.
4. The authors determined the crystal structure of SciW L66A in complex with its substrate, while the functional assays in Figure 6 were conducted using EagT6. Given the clear differences between the two proteins, additional cell-based functional validation experiments using the SciW mutant should be included to support the structural findings.
5. The dissociation rates between SciW-TMD and EagT6-TMD appear to differ significantly, suggesting that their dissociation mechanisms may not be identical. Therefore, the intermediate dissociation state of SciW-TMD captured in this study may not be universally representative. It would be valuable for the authors to further elaborate on this point in the discussion, addressing the potential differences in dissociation mechanisms and the broader relevance of the observed intermediate.

Minor concerns:

1. In Figure 6B, the expression levels of Tse protein are analyzed, and the authors suggest that different mutations in Eag affect bacterial competition activity by altering Tse protein stability. However, the figure lacks a quantitative analysis of Tse protein expression. From the data, the Tse protein expression in the L66A group is lower than in the Q102A group, yet the L66A group shows higher competition activity. The authors should provide a quantitative analysis or offer an alternative explanation for this discrepancy.
2. There is an extra scale bar in the lower right corner of the eagT6 label in Figure 6B. It is recommended to remove this unnecessary scale bar.

3.The panel numbers (A, B, C) in different figures are inconsistently sized. It is recommended to adjust them to a uniform size for better presentation.

4.For the Eag variants, the authors only performed unfolding assays to assess their resistance to chemical denaturation. I recommend that the revised manuscript include NanoDSF measurements to evaluate the stability of the Eag variants both individually and in complex.

5.In Figure S6H, the unfolding rate of the bound state of the EagT6 L66A mutant appears to be faster than that of the unbound state. This seems counterintuitive. Is the TMD still bound to the EagT6 L66A protein under these conditions? Further clarification or experimental validation may be needed to confirm the binding status.

Version 1:

Reviewer comments:

Reviewer #1

(Remarks to the Author)

This is a paper that I earlier reviewed and I have to commend the authors to address all my points. These were addressed by including points of discussion and updated figures (e.g fig.8) as well as experimental data whenever feasible (e.g Q106A in SciW). The authors also further detailed the importance of their work the advance the T6SS field.

Reviewer #3

(Remarks to the Author)

The authors have replied to all comments and made several changes, mostly to the discussion and explanation of the results. This has made different parts of the manuscript clearer and more accessible.

The main question that remains open to me, related to the previous major point 2, is how the delta-G values and binding energy measurements, calculated from chemical and thermal denaturation, relate to the physiological events causing the dissociation of the interacting proteins, which are most likely based on mechanical transitions. In other words, can the chemical / thermal denaturation experiments give relevant insights into the events the authors aim to study? If yes, this connection (and possible limitations) needs to be pointed out more clearly.

In addition, the previous minor point 10 relating to the competition assay shown in Figure 6 has not been addressed in my view. This is the only experiment that directly links the results of the study to physiological effects in the bacteria, and the results do not match. This is not a detail, but a core result. Downplaying these strong and statistically highly significant changes as "very subtle" and "details" does not appear appropriate. The authors claim to have "clarified the text" in this respect, but it is not clear where and how. This central point needs to be addressed.

Reviewer #4

(Remarks to the Author)

Reviewer #5

(Remarks to the Author)

I don't have further question.

Reviewer #1 (Remarks to the Author):

>The bacterial type VI secretion system (T6SS) is a potent antimicrobial weapon used by Gram-negative species. The T6SS can deliver numerous toxins with a broad range of biochemical activities, including enzymatic such as peptidoglycan hydrolases or lipases, as well as membrane pore forming. Targeting and loading of effectors onto the T6SS machine can involve a range of chaperones of which the Eag family has been involved in protecting hydrophobic and transmembrane domain (TMD) regions in toxins, such as nicely described between EagT6 and the Pseudomonas aeruginosa toxin Tse6.

Here the authors studied in very fine detailed the molecular interactions between Eag and two T6SS effectors, T6SS and SciW from Salmonella Typhimurium. They performed in depth thermodynamic and biophysical characterization to monitor the high stability of the Eag-TMD interaction that prevents transmembrane domains to prematurely insert into membranes as well as the subsequent effective release of this interaction. To support their data, they investigate structural features of domain interactions and identify key residues essential for the stability of the complex and which are found in a knob-in-hole packing arrangement features. This is further validated by genetic studies in vivo demonstrating that lack of stability in the Eag-TMD by substituting these key residues results in absence of killing during interbacterial competition.

Overall, the study is very technical and methodological, but very convincing although the gain in knowledge is rather incremental

Response: We thank the reviewer for their time and suggestions to improve our manuscript, especially their positive comments on the rigor and high quality of our experimentation. With our manuscript edits, we hope we have more clearly emphasized the significance and broad interest of our work. Including the methodological impact to study interactions with integral membrane proteins.

Specific comments

>The difference in lifetime and stability between the complex EagT6-Tse6 and Eag-SciW is astonishing, 18s vs 15h, respectively. Although the authors put these two complexes in different classes, it seems that the key residues involved in stability are the same in

both complexes. Could this mean that other domains in the T6SS effector may still strongly influence the stability of the complex and possibly that additional partners (line 486) strengthen and weaken the interaction. For example, if Eag-SciW1 separation requires disruption of more than one contact (lines 369-370) then this may be catalyzed by additional partner. Do the Eags have other partners that could be identified for example through in vivo pull-down.

Response: We agree with the reviewer that another binding partner likely catalyzes release. Our work details the key interactions and conformational changes of the Eag required to bind/release the TMD. In the discussion we hypothesize that another T6SS related protein is required to induce a conformational change in the Eag. Our results support that the conformational change includes moving L66 away from the TMD to initiate TMD dissociation. We have emphasized this to make it clearer for the reader. This includes edits to Figure 7 showing that L66 is part of the native conformational change upon TMD binding.

Attempts to discover additional Eag binding partners has been an ongoing line of study in our lab and has already been previously attempted. Notably, pull-down experiments with Eags have already been performed in *Pseudomonas* but only revealed the interaction with the cognate effector Tse6 and the VgrG.

Whitney et. al. Cell. 2015 Oct 22;163(3):607-19. doi: 10.1016/j.cell.2015.09.027. Epub 2015 Oct 8. PMID:26456113

Given that the previous pulldown results only found (Eag-TMDeffector-VgrG) any additional binding partner is likely to be transient which required challenging timing and/or advanced cross-linking to capture in a pull-down.

Even after this complex optimization, potential hits must be cloned and purified to test binding with Eag and Eag-TMD complexes. Mass-spectrometry identification of proteins from pull-downs commonly gives a list with many false-positives, and optimization of proteins for over-expression for binding assays is non-trivial. We have not yet been able to prove another Eag-T6SS interaction. Regardless, the results of a binding-partner discovery study when complete would constitute an entire manuscript of work on its own.

However, in the discussion we have stated that additional experimentation would be required to discover other Eag T6SS protein interactions.

>Are there mutations that could be identified that would dramatically increase the lifetime of Eag-Tse6 or dramatically decrease lifetime of Eag-SciW1.

Response: We agree that this is an interesting question. We do note that we have already discovered mutations that dramatically decrease the lifetime of SciW-Rhs1 (Q58A and L66A), albeit not to the level of Eag-Tse6.

However, creating gain-of-function mutations in any protein is non-trivial even with modern programs (RFdiffusion etc.). Furthermore, the aim of the current work is to determine the native molecular features of the chaperones required for effector TMD binding and dissociation. A study to redesign the Eags for increased TMD binding by introducing artificial stabilizing interactions is beyond the scope of the study.

>The authors suggest that the contact disruption within the Eag complexes occur by conformational change once the effector is in the T6SS baseplate (line 472). The firing and contraction mechanism associated with T6SS dynamic, involves a drastic conformation change in the baseplate from dome to starshaped. Would this contraction event contributing to the energy requires to disrupt Eag complexes. Could the Eag complex loaded in the T6SS nanomachine be modelled in the context of an extended or contracted version of the T6SS?

Response: We agree with the reviewer. A mechanism of Eag-TMD dissociation could be that a T6SS baseplate protein binds the Eag and the conformational change of the baseplate is tied to removal. The energy could be used to induce an Eag conformational change at the time of contraction to dissociate the TMD. Our data shows that the wild-type Eag-TMD complexes are incredibly stable, but the mechanical stress of T6SS firing would likely provide enough energy to break the complex (PMID: 25822993). Or at least provide enough energy to induce a conformational change that disrupts the interaction with L66 and/or Q58. We have added this as a potential model (discussion, Figure 8).

>In figure 6 the most drastic impact for interbacterial competition is the Q102A substitution which is at least as impactful than the L66A substitution. The equivalent of Q102A substitution in the Eag-SciW1 complex was unsuccessful (lines 231-232). As such there is very little data or discussion on that particular residue in the main text. Yet could it be a difference that may contribute explaining the lifetime differences between Eag-Tse6 and Eag-SciW1. This may be worth to investigate and discuss further.

Response: We agree with the reviewer. As we were unable to make Q106A in SciW we have ordered it from Genscript. The results of these assays have been added to all other SciW variant figures (Figure 3, Table S2, Table S3, Table S5, and Figures S4-7).

Q106A had little to no effect on the interaction between SciW and Q106A. This point has been added to the discussion of chaperone mechanistic differences.

>Overall, the study is very technical and methodological, but very convincing although the gain in knowledge is rather incremental

Response: We thank the reviewer for the positive review but would like to stress and clarify the broad importance and impact of our work.

First, our paper provides the only mechanistic details and model for how Eag chaperones are removed from effectors as part of the T6SS secretion process. To our knowledge, our work is currently the only study that provides energetic and detailed mechanistic data of how any T6SS chaperone is dissociated from its cargo. Because of this, our biophysical data is a significant step forward in the field of T6SS.

Second, a significant scientific advancement of our work was the development of an experimental strategy and model (eq. 1-9) to overcome the obstacles of studying the interaction of a soluble chaperone and an insoluble, unstable membrane protein. Specifically, because the TMD can only be produced already bound to the Eag. We succeeded and have extracted detailed mechanistic binding/dissociation data on Eags and their effectors that was previously unattainable.

Third, our findings are of interest not only for T6SS chaperones, but in the field of protein chaperones in general. This is in part because Eags are chaperones that unfold cargo and stabilize a mis-folded conformation to inhibit native folding until the proper time. The mechanistic and energetic details of how the Eag accomplishes this are relevant to understand other secretion chaperones and general cellular chaperones that work to unfold and then refold proteins.

Finally, our experimental approach can also be a model system for other groups studying chaperones or proteins that require co-expression with a binding partner for solubility, especially membrane protein chaperones.

Reviewer #2 (Remarks to the Author):

>Since my experience with the various biophysical techniques used in this manuscript is limited, I will only comment on the phenotypes related to T6SS function and impact of this paper on the field. This is overall a well written manuscript that addresses an important aspect of T6SS effector loading. The insights are interesting and such a detailed analysis is important for the field. I see no missing controls for the experiments testing phenotypes of the mutations generated based on the biochemical analysis. I suggest to add a figure with a cartoon like model for effector loading on the T6SS spike and removal of the chaperon (synthesis of the presented data and previous knowledge). This would help readers to imagine the whole sequence of events during T6SS assembly and effector loading.

Response: We thank the reviewer for their positive assessment of our work, the writing of the manuscript, and the significance of our findings. We have added a mechanistic figure as suggested (Figure 8).

Reviewer #3 (Remarks to the Author):

>Effectors of the type VI secretion system (T6SS) are usually delivered with the ejected needle (Hcp) or its tip (VgrG/PAAR), either as direct fusions to these proteins or bound

to them. Some effectors require chaperone proteins that can take various roles, but are usually required for delivery of their cognate effector, without being exported themselves. One such class of chaperones are the Eag chaperones, which in particular bind transmembrane domain (TMD) effectors. The authors of this manuscript had investigated the binding of Eag chaperones to a conserved motif found in TMD-containing effectors in eLife in 2020. In this manuscript, they follow up on this publication by addressing the obvious question how the Eag chaperones are removed from the effectors prior or during secretion. To this aim, the authors measure the stabilization of two Eag chaperones by the presence of the TMD of their respective effectors. They complement this analysis by structural analysis and a set of competition and expression experiments on a set of mutants.

The manuscript investigates an interesting question and the results appear solid and are well-presented (excluding the points raised in the minor comments). However, the reasoning and logical connections are hard to follow in various places, and it is unclear if the experiments (largely unfolding by chaotropic agents) allow for the conclusions on the T6SS assembly or export process. It is, therefore, difficult to assess whether the authors' model and conclusions are justified by the data.

Response: We thank the reviewer for their overall positive assessment of our work and their suggestions to improve our study. We have made edits to both improve readability and to make the reasoning behind our line of investigation clear to the reader.

Major comments:

>1. In particular, it remains unclear at which time point the authors think the Eag chaperones are removed from the TMH effectors. The scheme in the authors' 2020 eLife paper shows the Eag chaperones as parts of the assembled T6SS prior to firing. This assumption is also retained throughout the manuscript, until, suddenly, in the description of the model (lines 463 onwards), Eag-TMD release is hypothesized to occur upon binding to the baseplate (which is not analyzed or even mentioned once in the results).

Response: We apologize as we did not intend to suggest a model at odds with our past publication. The exact words used were:

“Next, we hypothesize that the Eag is removed from the TMD within the baseplate/membrane complex. ... Within the baseplate, a conformational change in Eag is induced to disrupt the Eag-TMD complex.”

To agree with our past publication, we only meant to suggest that dissociation occurs within the baseplate. We currently do not know the exact time of removal or if the Eag-TMD does in fact directly bind another T6SS protein for dissociation. As also addressed for reviewer 1, pull-down experiments to find and subsequently prove another Eag interaction partner have been tried but have yet to produce substantial results.

We have further clarified this point in the model discussion section for the reader.

>The two scenarios are vastly different, as removal upon firing would likely imply mechanical stress, which is quite different from the scenario to which the measurements likely apply (GdmCl-induced unfolding).

Response: We would again like to emphasize that although we used chaotropic agents to study Eag-TMD unfolding and stability, **all the thermodynamic and kinetic parameters reported in table 1 are native values in buffer without any denaturant. Our measurements and model yield free-energy and kinetic values in a standard *in vitro* physiologically relevant experimental buffer.**

By studying Eag and Eag-TMD folding at different concentrations of chaotropic agents we can then fit the data with our model (equations 1-9) and extrapolate to 0 M GdmHCl. None of the reported binding or dissociation values are influenced by GdmHCl.

The data shows how much energy is required and the molecular contacts that are critical for Eag-TMD complex formation and thus are likely disrupted to promote TMD dissociation. The energy for disruption and TMD dissociation may then come from the mechanical stress of T6SS firing (PMID: 25822993).

Additionally, the biophysical data are supported by our current SciWL66A structure that demonstrates a partial release of the TMD. Furthermore, our previous X-ray structures show that the region containing L66 is part of the Eag conformational change induced upon TMD binding for both EagT6 and SciW. Thus, a hypothesis that disruption of the interaction of L66 with the TMD is part of the dissociation mechanism is supported by our previous structural data and all mechanistic data presented in this manuscript.

However, expanding upon the details of a firing-coupled model where mechanical stress may help provide the energy for Eag removal was also suggested by reviewer 1, and we have expanded our discussion.

We have also included additional figure panels illustrating known Eag conformational changes from our various structures in Figure 7. Also as suggested by reviewer 2, we have added dissociation model in Figure 8.

>2. The authors state that they have been unable to express TMD effectors without their chaperones and understandably, therefore, choose indirect methods to analyze the binding and unbinding. However, it is not immediately obvious that these methods allow to draw the conclusions made in the manuscript. To the best of my knowledge, changes in the T_m provide qualitative, but not quantitative binding affinities. Perhaps more importantly, this applies to equilibrium kinetics, but not to mechanical stress, as possible relevant for the removal of the T6SS chaperones. If the methods do allow for these conclusions, the underlying reasoning needs to be made clearer to the reader.

Response: We have not used T_m s (melting temperatures) for the derivation of our thermodynamic and kinetic parameters. All ΔG and rate constants ($k_{\text{unfolding}}$) are

measured from chemical denaturation then extrapolated to physiological buffer using our model in equations 1-9 (PMID: 11013396).

All ΔG° unfolding, ΔG° binding and k° unfolding in table 1 are native values in a physiologically relevant buffer without any denaturant present.

To make this clear to the reader, we now state this immediately in the first chemical denaturation section:

“We next probed Eag chaperone resistance to chemical denaturation. By measuring Eag and Eag-TMD unfolding by denaturants, we will be able determine the free-energy (ΔG) of Eag stabilization and the binding energy of the Eag-TMD interaction. Specifically, the ΔG of unfolding measured at various denaturant concentrations is used for extrapolation to 0 M denaturant. This yields native denaturant-independent ΔG s of folding and TMD binding energy in a physiologically relevant buffer (PMID: 11013396).”

Additionally, we have edited the manuscript in other locations to also better explain to the reader how and why our thermodynamic and kinetic parameters are physiologically relevant.

We do agree with the reviewer that the chemical denaturation binding calculations can be viewed as estimates as they depend upon the assumptions in our model (equations 1-9). We have clearly stated the assumptions inherent in the model for the reader:

“The effector TMD binding energy (ΔG° binding) and association constant (K_{TMD}) can then be obtained from comparing Eag ΔG° unfolding to Eag-TMD ΔG° unfolding. This calculation requires the following assumptions: 1) the TMD is a small ligand compared to the Eag, 2) TMD binding does not significantly alter the Eag fold, 3) all free TMD in solution results from complex dissociation, 4) the chaperone folds by a 2-state model, and 5) the TMD is assumed to selectively bind the folded form of the Eag with an association constant K_{TMD} .”

However, we note that our mechanistic details are based on the differences in energy between Eag, Eag-TMD, and the point variants. As we have made the same assumptions for all calculations, any error inherent in these simplifications is subtracted.

We also note that we have stated the binding values are estimates:

“This directly relates the change in Eag unfolding standard free energy to that of TMD binding at native conditions. The unfolding data of Table 1 allow us to estimate a ΔG° binding of approximately -176 kJ/mol for the SciW-TMD complex and a ΔG° binding of -116 kJ/mol for the EagT6-TMD complex.”

>3. *Negative controls, such as the addition of generic TM or of the respective non-cognate TMD, are missing for the key experiments.*

Response: We again emphasize that the TMDs are unstable and insoluble unless they are co-expressed already bound and stabilized by the Eag (PMIDs: 33320089, 30177742, 26456113 and current Figure 6). We cannot add a purified TMD to a purified Eag. These facts are why we pursued denaturation of Eag-TMD complexes to extract dissociation data. The suggested experiments are simply not possible given the system.

Additionally, we strongly disagree that suggested experiments with a generic TM or a non-cognate TMD are appropriate controls. Comparison of Eag binding to another protein will neither prove or disprove effector binding or the measured binding energies of the Eags to their known and proven cognate TMDs.

Supporting our work, there is extensive data and proof showing that Eags bind their cognate effector TMD and are specific to their effector class (Ahmad et. al, *elife* 2020 Figures 1-5 and associated supplemental figures PMID: 33320089). Several other publications show similar findings (PMIDs: 30177742, 26456113, 33323487, 34853317).

However, we have provided extensive controls for Eag-TMD binding and to support our results. First, the binding controls are internal for this study. Namely, Eag-TMD complexes relative to Eag alone. And subsequently, wild-type proteins compared to point-variants. Further controls include TMD-affinity tagged purifications that select for Eag-TMD complexes, DLS that shows the variants are properly folded, and additional nanoDSF data (Table S2, Figure S5 and Figure S6). These all show that we have in fact purified the relevant Eag-TMD complexes of wild-type and point-variants.

Minor comments:

>1. *Why does Rhs1(1-59) aggregate on SDS-PAGE gels (Fig. S1 and S5), but not in the DLS measurements (Fig. S5)? Tse6(1-61) behaves different in this respect.*

Response: On denaturing SDS-PAGE gels membrane proteins (like the TMDs) sometimes do not migrate as expected. This is due to their highly hydrophobic nature and interaction with SDS micelles. Alpha-helical membrane proteins (like the TMDs) often aggregate or oligomerize on SDS-page gels. The reason that the Tse6-TMD runs as expected could be due to sequence differences relative to the Rhs1-TMD that might promote oligomerization. These observations are usually empirical and are difficult to predict.

Detergent binding explains anomalous SDS-PAGE migration of membrane proteins Proc Natl Acad Sci U S A. 2009 Feb 10;106(6):1760-5. doi: 10.1073/pnas.0813167106 PMID: 19181854

SDS-induced oligomerization of Lys49-phospholipase A2 from snake venom

The reason no TMD aggregates are observed by DLS is because:

- 1) The material is co-purified SciW-Rhs1TMD. The Rhs1-TMD is tightly bound to SciW. Our binding data indicates antibody level affinity (at least pico molar). When it is bound to SciW, the TMD is packed within the SciW claw and stabilized. There are likely no free TMDs above the detection threshold.
- 2) If any TMD has separated from SciW it may have precipitated or degraded (PMIDs: 30177742, 26456113). All DLS samples must be centrifugally filtered.

> *Additional controls, beyond the fact that Rhs1(1-59) still binds to SciW, would provide evidence for its native binding, and the respective conclusions.*

Response: Our past and current work has provided extensive data to prove that SciW natively binds Rhs1(1-59).

- 1) SciW and Rhs1 (full-length) binding was shown in our previous eLife paper by pull-down (Figure 1 in eLife PMID: 33320089).
- 2) In the previous eLife paper, SciW-Rhs1(1-59) was purified by metal affinity (TMD His-tagged) followed by size exclusion chromatography (SEC). As only the TMD is selective for Ni-NTA column binding, we will only purify SciW if SciW is tightly bound to the TMD. Additionally, SEC co-elution requires stable complexes.
- 3) We have solved a co-crystal structure of SciW bound to Rhs1(1-59) (6XRR). To capture a co-crystal structure between proteins requires specific native binding.
- 4) Purifications of all Eag-TMD complexes in the current work also use nickel affinity chromatography followed by SEC where only the TMD is His₆-tagged. This specifically selects for the SciW-Rhs1(1-59) complex. This is a strong control for native binding. (Figure S5)
- 5) All biophysical experiments of SciW-Rhs1 (1-59) and point variants show a drastic increase in stability relative to apo-SciW. This includes Tms by nanoDSF and DSC, chemical denaturation experiments, and stopped-flow measurements. Note that the nanoDSF and DSC are in a standard physiological buffer. (Figures 1-2, 45 and Figures S6)
- 6) We have shown in this study that the binding affinity of SciW to Rhs1 is at least pico molar. This is indicative of a specific biologically relevant complex.
- 7) We have solved an X-ray crystal structure of a SciW point variant bound to the TMD of Rhs1. (Figure 7). Even weakening the binding affinity still allows for the capture of the TMD bound to SciW.

Additionally, there are many examples in the literature that show that the adjacently encoded Eag and Rhs effector are native binding partners.

1. Elife. 2020 Dec 15;9:e62816. doi: 10.7554/eLife.62816 PMID: 33320089
2. Proc Natl Acad Sci U S A 2020 Dec 29;117(52):33540-33548. doi: 10.1073/pnas.1919350117 PMID: 33323487

3. Nat Commun. 2021 Dec 1;12(1):6998. doi: 10.1038/s41467-021-27388-0. PMID: 34853317
4. PLoS Pathog. 2022 Jan 5;18(1):e1010182. doi: 10.1371/journal.ppat.1010182 PMID: 34986192

>2. *Is there a difference between chaperones and adaptors? If so, it should be explained, if not, this should be mentioned when mentioning adaptors (e.g. Tap adaptors).*

Response: Yes, there is a difference. We have clarified this in the introduction text:

“A critical step in cargo effector secretion is the selective recruitment or piloting of effectors to the T6SS by accessory proteins that act as chaperones or adaptors. Chaperones stabilize effectors in a “secretion-competent” state and affect protein folding, whereas adaptors typically only aid in the binding of effectors to T6SS machinery. The terms chaperones and adaptors are often used interchangeably in T6SS literature, especially as some chaperones also aid in effector loading (PMID:34092034, 33320089). Despite the ambiguity, chaperones and adaptors are essential for the proper delivery and toxicity of numerous T6SS effectors.”

>3. *The reason to focus on the first TMD of Tse6 (or a reference showing binding of this TMD only to EagT6) should be provided.*

Response: The reason we focus on TMD1 is that we lack a crystal structure of EagT6-TMD2. Additionally, the AlphaFold3 prediction poorly predicts the conformation of the TMD. The highest confidence region of the TMD only ranges from 65-80 pLDDT with most of the TMD in the 0-50 range (Figure R1). Given this, we cannot make any detailed mechanistic conclusions about EagT6 and TMD2.

FigureR1: AlphaFold3 model of EagT6 and putative TMD2 region residues 167-222

We have made this clear to the reader in the discussion:

“It is important to note that we lack a crystal structure of EagT6-TMD2 and AlphaFold3 fails to predict a confident model (pLDDT < 65 for TMD2). Our mechanistic dissociation data currently only applies to EagT6-TMD1. However, L66A is sufficient to inhibit secretion *in vivo* strongly supporting our TMD1 dissociation model and importance of TMD1 binding.”

>4. *Lines 66/67: This statement does not apply to all effectors, and is therefore misleading in its current state.*

Response: We have changed the language from ‘cargo effectors’ to be ‘for many effectors’ to be more general.

>5. *The role of chaperones should be explained more clearly – lines 67-69 are rather vague and unspecific.*

Response: We have clarified the differences between chaperones and adaptors and added that chaperones typically affect protein folding. Additionally, we provide several examples of known chaperones and adaptors for detail.

>6. *Line 123: Rhs should be called Rhs1, as elsewhere in the*

manuscript. **Response:** Thank you, these have been corrected.

>7. *The dimer formation of Eag chaperones, one of their defining features, is not taken into consideration in the denaturation experiments (Fig. 1, 2), although likely to influence the results. Likewise, it is not taken into account in the equilibrium calculations described in lines 692-698. The authors seem to assume an essentially complete dimerization of the chaperones, but no evidence for this is provided.*

Response: We agree with the reviewer. We need to provide data demonstrating the oligomeric state of the Eags. In Figure S5 we performed extensive DLS on all Eag samples in the study.

We have added the wild-type DLS curves and measurements to Figure S1. Additionally, we have added a table listing the measured particle size and polydispersity index (PDI) of wild-type and all point variants in Table S2.

As demonstrated Eag and Eag-TMDs have a radius of ~2.8 to 3.2 nm, which based on the crystal structure is the correct radius for a dimer. This is also in agreement with the accepted sizes of globular protein standards that indicates a MW of near 35 kDa, or a dimer.

We also note that the TMD binding site is within the pocket of the claw and thus will not significantly increase the size of the protein in solution. Additionally, we only observe one type of particle in solution shown by one primary peak and a low PDI. The crystal structures and cryoEM data of Eags show that Eags are dimers when bound to the TMD or when unbound, further supporting that Eags are exclusively dimers in solution.

Nat Microbiol. 2018 Oct;3(10):1142-1152. doi: 10.1038/s41564-018-0238-z. PMID: 30177742

Elife. 2020 Dec 15;9:e62816. doi: 10.7554/eLife.62816 PMID: 33320089

Finally, we note that K_{unfolded} is defined in equation 1 as dimer to monomer:

The relationship between K_{unfolded} and free energy in equation 4 is then the basis for the derivation of all other equations:

$$\Delta G_{\text{unfolding}} = -RT \ln K_{\text{unfolding}}$$

>8. *How exactly does the difference in denaturation by GdmCl and Urea allow for the determination of the free energy of Eag-TMD unfolding (statement in lines 163-164)? Is urea used afterwards and what for?*

Response: Figure 2A shows that SciW-TMD is immune to denaturation by urea, so no folding information or free energy can be calculated. Instead, we found that GdmCl is sufficient to denature all Eags and complexes in the study. Urea is thus not used again in the study. We have clarified this for the reader.

>9. *Are lines 212-218 new results (as implied by the inclusion in the results part and lack of reference)? If they are instead a recapitulation of the results in ref. 27, they should be moved from the results part to the introduction, or this fact should be at least made clear.*

Response: Although, they are past results it is important to keep this information in the section for the reader to clarify the selection of point variants. We have clarified that these are already known results and further emphasized their origin to the eLife reference.

>0. *The cellular concentration of Tse6 (which the authors, conceivably, assume to be determined by the binding to its chaperone) does not correlate well with the competitive index. For example, the Q58A mutant, in which the level is similar to the WT, has a strongly decreased fitness compared to the WT, and S41A, which retains expression, is*

doing worse than I24F, where almost no Tse6 could be detected. Can the authors explain these discrepancies?

Response: We thank the reviewer for their attention to these details; however, this comment is an over-interpretation of the data presented in figure 6. The experiment presented in figure 6A demonstrates that any of the mutations introduced at the EagT6-Tse6 binding interface reduce the competitive fitness of the donor over the recipient relative to wild-type EagT6. We cannot make claims about the significance of these mutations relative to one another, as this competition assay lacks the statistical power to detect the differences between individual mutants, which are very subtle as the reviewer points out. The Western blot suggests that Q58A is less disruptive to the Eag-Tse6 complex than other mutations such as L66A, which is consistent with our biophysical data. We have clarified the text.

>1. What is meant by “the Eag chaperone catalyzes completion of the PAAR domain with the prePAAR motif” (lines 463-465, in the description of the model)? The expression is not contained in the listed ref. 27, and the mechanics described by it remain unclear.

Response: The Eag positioning the prePAAR motif to complete the PAAR domain fold was discussed and experimentally tested in the elife paper (Figure 6 and supplement). This mechanism is also included in the proposed model Figure 8 (elife PMID: 33320089).

We have made this clearer for the reader in the text.

>2. There is a certain inconsistency and confusion in the gene identifiers given by the authors (lines 547-552 and elsewhere):

- SL1344_0268, assigned to Salmonella SciW in line 547, encodes instead for TssC, the large subunit of the T6SS contractile sheath*
- SL1344_0286, assigned to SctW elsewhere, seems to encode for an RHS protein instead*
- SL1344_0285, assigned to Rhs1, encodes for a much smaller protein*
- Likewise, the two protein identifiers may have been mixed up for the Pseudomonas genes*

Response: We thank the reviewer for finding these typos. 0268 is a typo for 0286. 0285 is SciW and 0286 is Rhs1. They were mistakenly reversed in the text and tables. We have also fixed the mistaken PA identifiers. Furthermore, as much of the T6SS work in Salmonella has used the LT2 strain we note what the equivalent LT2 genes are to help clarify for the reader.

Reviewer #4 (Remarks to the Author):

>I co-reviewed this manuscript with one of the reviewers who provided the listed reports. This is part of the Nature Communications initiative to facilitate training in peer review

and to provide appropriate recognition for Early Career Researchers who co-review manuscripts.

Response: We are happy to help support ECR training and we thank the reviewer for their time.

Reviewer #5 (Remarks to the Author):

>The manuscript by Van Schepdael et al. explores the mechanism of effector transmembrane domain (TMD) release from Eag chaperones, crucial for Type VI secretion system (T6SS) function. The authors use a combination of biophysical techniques and bacterial assays to demonstrate that small perturbations in Eag-TMD interactions, such as disrupting hydrogen bonds or knob-in-hole packing, can trigger TMD release. Their work underscores the essential role of conformational changes in Eag chaperones for the dissociation of TMDs, a critical step for effector secretion and bacterial fitness. Additionally, the X-ray co-crystal structure of the Eag-TMD complex captures an intermediate state of TMD release, offering a novel perspective on this process. This study contributes significantly to our understanding of how T6SS chaperones regulate effector folding and secretion. The insights provided in this study are important and should be of broad interest. While, to strengthen the work, the authors may need to address my concerns listed below.

Response: We thank the reviewer for their overall positive assessment of our work. We have made both edits to the manuscript and performed additional experimentation as recommended.

Major concerns:

>1.The study investigates the mechanism of Eag chaperone dissociation from effector proteins, but the extent to which the in vitro-derived dissociation mechanism reflects the actual dissociation process in bacterial cells is questionable. For example, the manuscript primarily examines the impact of different mutations on bacterial secretion but does not confirm whether these mutations specifically affect Eag dissociation, which then influences bacterial secretion and survival. The authors should include in vitro dissociation kinetics experiments and in vivo/in-cell binding assays to demonstrate the role of chaperones in regulating effector protein folding and their involvement in T6SS under physiological conditions.

> The authors should include in vitro dissociation kinetics experiments... under physiological conditions

Response: We thank the reviewer, but the manuscript already explicitly addresses the reviewer's concerns. The presented data (Table 1, Figure 2-5, Figures S2-3, and Figures S8-S10) are the requested *in vitro* binding and dissociation kinetics.

The reported data in Table 1 are native free energies of folding, binding, and rate constants in physiological buffer independent of challenge with denaturant as requested. Figure 2G-H, Figure 4E-H and Figure 5 shows how the denaturation data was used to obtain the native values for wild-type Eags and the two variants of SciW and EagT6 in an *in vitro* physiologically relevant buffer (no denaturant).

Again, the TMDs are insoluble and degrade unless they are bound and stabilized by an Eag. It is not possible to add purified TMD to Eag in a physiological buffer then perform experiments such as ITC, MST, or BLI. We can only make Eag-TMD complexes for *in vitro* experiments. This limitation is why we denatured the Eag and Eag-TMDs then created a model to extrapolate to physiological conditions to obtain native *in vitro* binding energies and dissociation kinetics.

> vivo/in-cell binding assays to demonstrate the role of chaperones in regulating effector protein folding and their involvement in T6SS under physiological conditions

Response: We have already performed experiments to correlate our *in vitro* binding data to *in vivo* function. In Figure 6 we measure the effect each point variant has on the ability to stabilize Tse6 (which contains the TMD) (Figure 6B) and promote secretion of Tse6 to kill other cells (Figure 6A). These experiments were performed in the bacteria *Pseudomonas aeruginosa* that natively carries the T6SS apparatus, EagT6 and Tse6.

Additionally, significant past work has shown that EagT6 binds Tse6 *in vivo* and stabilizes the protein. We now show that are binding point variants can no longer bind and stabilize Tse6 as efficiently as wild-type and reduces T6SS mediated killing.

Cell. 2015 Oct 22;163(3):607-19. doi: 10.1016/j.cell.2015.09.027. PMID: 26456113

Nat Microbiol. 2018 Oct;3(10):1142-1152. doi: 10.1038/s41564-018-0238-z. PMID: 30177742

>2. The SciW-TMD L66A crystal structure presented in the study captures an intermediate state of Eag chaperone dissociation from TMD and supports the proposed dissociation model. However, this structure was obtained by introducing the L66A mutation to stabilize the dissociation intermediate. It is unclear whether this structure fully reflects the dissociation process under physiological conditions in bacterial T6SS, as wild-type chaperones in cells may dissociate differently. Could the structure of the wild-type chaperone dissociating from TMD under natural conditions be obtained by optimizing parameters such as temperature?

Response: It is near impossible to predict how a protein will crystallize or what conformation will be captured. It is also near impossible to alter crystallization conditions to selectively force a protein into a different conformation. Typically, changing crystallization conditions from the known buffer results in no crystals. For example, to

obtain the structures we have previously published and present here required screening hundreds of conditions (including temperature) to even acquire diffracting crystals.

However, to help address the reviewer's concern of relevance we note that the region of the Eag that changes conformation upon binding the TMD includes residue L66. Specifically, in both SciW and EagT6 residue L66 changes conformation moving over the TMD to make the critical knob-in-hole interactions (elife PMID: 33320089). We have added an overlay of bound and unbound Eag structures highlighting the movement of L66 upon binding in the wild-type proteins to Figure 7. It should be clearer to the reader that this comparison shows how our point variant structure disrupts this essential interaction and partially reverses the induced L66 conformational change upon binding. Furthermore, L66 is highly important for TMD binding in all our interaction studies. Thus, we believe it is a reasonable hypothesis that the structure represents a model of an intermediate of TMD dissociation where the L66 knob-in-hole has been disrupted.

>3. The study attributes the dissociation pathways of Class 1 (SciW-TMD) and Class 2 (EagT6-TMD) effector proteins to “the different roles of Q58/L66 contacts” and “structural differences in transition states.” However, the TMD sequences, overall conformations, and coupling with VgrG/PAAR differ significantly between the two effector classes, suggesting that other factors may regulate chaperone dissociation. The authors should include further experiments to confirm that the chaperone-TMD interactions are responsible, rather than other factors playing a key role in regulating T6SS.

Response: We agree that other factors regulate chaperone dissociation. This was outlined in the discussion of the manuscript. The potential models are that a T6SS ATPase in the baseplate catalyzes dissociation or that a baseplate protein binds the Eag and forces the TMD out of the Eag coupled to T6SS firing (added in the revision). A model schematic has also been included in Figure 8.

At no point do we propose that the Eag can self-catalyze the dissociation of the TMD. Our biophysical data provides a model for how the Eag can be manipulated to release the TMD. We show what contacts are critical to TMD binding, show that they linked to the conformation change that the Eag undergoes when it binds the TMD, and hypothesize another protein and/or force manipulates the Eag to disrupt the interaction of L66 and Q58 with the TMD. This then results in TMD dissociation.

Furthermore, attempts to discover Eag binding partners have been performed in *Pseudomonas* but only found the EagT6-Tse6-VgrG complex (PMID: 26456113). We have attempted to find additional Eag partners but have been so far unsuccessful. Any binding partner that catalyzes Eag-TMD dissociation is likely to be transient which requires challenging timing and/or advanced cross-linking to capture in a pull-down.

Potential hits must be cloned and purified to test binding with Eag and Eag-TMD complexes which typically include many false-positives. Regardless, the results of a

binding-partner discovery study when complete would constitute an entire manuscript of work on its own.

>4. The authors determined the crystal structure of SciW L66A in complex with its substrate, while the functional assays in Figure 6 were conducted using EagT6. Given the clear differences between the two proteins, additional cell-based functional validation experiments using the SciW mutant should be included to support the structural findings.

Response: We agree with the reviewer that additional assays in *Salmonella* Typhimurium would enhance the current study. We have performed bacterial competition assays in *S. Typhimurium* with wild-type, a */iclP* deletion (T6SS negative control), and a */isciW* deletion (Figure S11). Although we are able to show T6SS dependent killing by *S. Typhimurium*, the */isciW* deletion only shows an intermediate phenotype. We note that this is in agreement with past results showing that deletion of *sciW* has a limited effect on bacterial fitness in a mouse model (PMID: 22493086).

Although wild-type does have a significant killing phenotype relative to the */iclP* deletion, the overall phenotype for *Salmonella* is very low compared to *Pseudomonas* and is barely significant from the */isciW* mutant (~1.6x ratio wild-type vs ~1.3x ratio */isciW*, p-value 0.035). Given this and that the *SciW* variants do not drastically disrupt TMD binding it is highly unlikely that any statistically significant differences in function can be detected between the wild-type and *SciW* point variants. Compounding this is the high experimental variability due to bile-salt induction (see error bars in Figure S11). As such we have not further pursued experiments in *S. Typhimurium*, but we have included the obtained data as supplemental Figure S11.

>5. The dissociation rates between SciW-TMD and EagT6-TMD appear to differ significantly, suggesting that their dissociation mechanisms may not be identical. Therefore, the intermediate dissociation state of SciW-TMD captured in this study may not be universally representative. It would be valuable for the authors to further elaborate on this point in the discussion, addressing the potential differences in dissociation mechanisms and the broader relevance of the observed intermediate.

Response: We agree. We have included additional dissociation mechanistic models as suggested. We had originally explicitly stated that we believe the mechanisms are different in the discussion, but we have made this clearer and expanded.

Minor concerns:

>1. In Figure 6B, the expression levels of Tse protein are analyzed, and the authors suggest that different mutations in Eag affect bacterial competition activity by altering Tse protein stability. However, the figure lacks a quantitative analysis of Tse protein expression. From the data, the Tse protein expression in the L66A group is lower than in the Q102A group, yet the L66A group shows higher competition activity. The authors

should provide a quantitative analysis or offer an alternative explanation for this discrepancy.

Response: As discussed in our response to reviewer 3 above, we are not drawing conclusions regarding the relative contributions of different mutants to the function of EagT6 *in vivo* - we are simply concluding that each of the mutations we introduced at the EagT6-Tse6 binding interface disrupt the competitive fitness of the donor strain against the recipient strain relative to wild-type EagT6. As evidenced by the overlapping error bars between the L66A and Q102A group, the competition assay lacks the statistical power to detect potential differences between these two mutations, which are very subtle if they exist at all.

>2. There is an extra scale bar in the lower right corner of the eagT6 label in Figure 6B. It is recommended to remove this unnecessary scale bar.

Response: The figure has been fixed.

>3. The panel numbers (A, B, C) in different figures are inconsistently sized. It is recommended to adjust them to a uniform size for better presentation.

Response: We thank the reviewer for spotting this to be corrected. We have made sure that our figures have all been made in the same professional graphics program with consistent font sizes. They have been designed to export as 2-column or 1-column of the appropriate size. We hope that automatic figure resizing by placing figures in Word or other programs has not re-created the problem.

>4. For the Eag variants, the authors only performed unfolding assays to assess their resistance to chemical denaturation. I recommend that the revised manuscript include NanoDSF measurements to evaluate the stability of the Eag variants both individually and in complex.

Response: These have been added to the supplemental as Figure S6. The data shows a similar pattern of Eag and Eag-TMD stability changes as chemical denaturation. However, unlike chemical denaturation we cannot derive binding energies from the nanoDSF which is why we did not originally include them.

>5. In Figure S6H, the unfolding rate of the bound state of the EagT6 L66A mutant appears to be faster than that of the unbound state. This seems counterintuitive. Is the TMD still bound to the EagT6 L66A protein under these conditions? Further clarification or experimental validation may be needed to confirm the binding status.

Response: We respectfully point out that this analysis is incorrect. Figure S6H is the steady state denaturation (not unfolding rate) using fixed 330/350 ratios. Additionally, the data for both the apo-EagT6 L66A and EagT6-TMD L66A are within experimental error. Furthermore, this experiment is repeated in Figure 4D using wavelength maxima

measurements as outline in the text. As clearly shown the apoEagT6L66A and TMD bound are essentially the same. The TMD-bound is not lower.

We demonstrate clearly in Figure S5 that EagL66A-TMD co-purifies, and thus the TMD is bound. The requested nanoDSF also shows a T_m shift for EagL66A-TMD relative to apo-EagT6 but interestingly shows the appearance of a shoulder at the apo-T_m (Figure S6H). This indicates that the TMD is likely dissociating from the Eag. We state clearly in the text that we believe the TMD has dissociated prior to our unfolding experiments:

“This indicates that minimal TMD binding to the EagT6 L66A variant exists under our assay conditions. Our fluorescence experiments are performed under dilute conditions and in the absence of crowding agents, it is likely that the TMD has dissociated prior to the assay.”

Response to review 2

We thank the reviewers for their positive response to our revisions and for their time to help us improve our work.

REVIEWERS' COMMENTS

>Reviewer #1 (Remarks to the Author):

This is a paper that I earlier reviewed and I have to commend the authors to address all my points. These were addressed by including points of discussion and updated figures (e.g fig.8) as well as experimental data whenever feasible (e.g Q106A in SciW). The authors also further detailed the importance of their work the advance the T6SS field.

Response: We thank the reviewer for their time and help, and we appreciate that we have addressed all their concerns.

>Reviewer #3 (Remarks to the Author):

The authors have replied to all comments and made several changes, mostly to the discussion and explanation of the results. This has made different parts of the manuscript clearer and more accessible.

Response: We thank the reviewer for their time and positive assessment of our revisions.

The main question that remains open to me, related to the previous major point 2, is how the delta-G values and binding energy measurements, calculated from chemical and thermal denaturation, relate to the physiological events causing the dissociation of the interacting proteins, which are most likely based on mechanical transitions. In other words, can the chemical / thermal denaturation experiments give relevant insights into the events the authors aim to study? If yes, this connection (and possible limitations) needs to be pointed out more clearly.

Response: As explained throughout the manuscript and in detail in our previous response to review, we used chemical denaturation as a tool to obtain the **physiologically relevant binding and dissociation energies and rates for the Eag-TMD complexes**. The thermodynamic and kinetic parameters reported in table 1 ($\Delta G^{\circ}_{\text{unfolding}}$, $\Delta G^{\circ}_{\text{binding}}$ $k^{\circ}_{\text{unfolding}}$) are obtained from extrapolating to 0 M denaturant concentration, and thus are the native physiological values. None of the reported values depend upon denaturant ($\Delta G^{\circ}_{\text{unfolding}}$, $\Delta G^{\circ}_{\text{binding}}$ $k^{\circ}_{\text{unfolding}}$).

The reported $\Delta G^{\circ}_{\text{binding}}$ (energy of TMD binding) is the **physiological energy** (as if we had used conventional methods such as ITC, MST, BLI) that the cell must overcome during the secretion process to remove the TMD from the Eag. The $\Delta G^{\circ}_{\text{binding}}$ for each variant allows to see how much energy that specific interaction contributed to the complex. We

also outline in detail the assumptions inherent in our model to calculate the physiological energy of Eag-TMD binding.

In the discussion we also highlight that previous studies have shown that T6SS firing produces enough energy to remove the TMD from the Eag. Furthermore, the sum of our biophysical, structural, and *in vivo* data shows that a mechanism to catalyze TMD release and lower the energy of Eag-TMD dissociation is to induce a conformational change in the Eag that moves L66 away from the TMD. These possibilities have been outlined in the discussion.

However, we have made further edits to clarify the significance of our data to the reader.

Introduction:

“Here, we measure the thermodynamic properties and unfolding kinetics of wild-type Eag-TMD complexes compared to several site-specific point variant Eag-TMD complexes. Our data yields the binding energies and dissociation kinetics of Eag-TMD complexes, which shows that although Eag-TMDs are remarkably stable only a small perturbation of the Eag is required to promote TMD dissociation.”

And we've also clarified in the secretion lines 196-213

“Specifically, the ΔG of unfolding measured at various denaturant concentrations is used for extrapolation to 0 M denaturant. This yields native denaturant-independent ΔG s of folding and TMD binding energy in a physiologically relevant buffer. Therefore, the data reveals the binding energy that must be overcome during secretion to remove the Eag.”

>In addition, the previous minor point 10 relating to the competition assay shown in Figure 6 has not been addressed in my view. This is the only experiment that directly links the results of the study to physiological effects in the bacteria, and the results do not match.

Response: We respectfully disagree with the reviewer in their assessment of the data. Specifically, the statement that ‘the results do not match’. The biophysical data and the *in vivo* data in fact do match and agree.

Notably, we quantify EagT6 and EagT6-TMD wild-type, the Q58A variant, and the L66A variant. In table 1 and data figures (2-5), show that relative to the wild-type (-116 kJ/mol), Q58A reduces the magnitude of the $\Delta G^{\circ}_{\text{binding}}$ (-60 kJ/mol) and L66A reduces the $\Delta G^{\circ}_{\text{binding}}$ further to a level that we cannot quantify in our assay (weaker than 1 μM Kd, or worse than -35 kJ/mol).

We state this explicitly in the text:

“This shows that the binding constant K_{TMD} for the L66A substitution has been reduced to less than $10^6 M^{-1}$ putting an upper limit of ~ -35 kJ/mol upon the $\Delta G^{\circ}_{binding}$.”

When the biophysical data (binding energies) of wt (-116), Q58A (-60), L66A (worse than -35) are compared to the *in vivo data* in Figure 6, we see exactly the same trend. A competition ratio of ~ 6 for wild-type, ~ 4 for Q58A, and ~ 2 for L66A. Additionally, all results have statistical significance relative to each other.

We again note that L66A does bind the TMD (Supplemental Figures 5-7) but at the dilute concentrations required for the quantitative denaturation assays the complex has already dissociated.

We also state this explicitly in the text:

“Our fluorescence experiments were performed under dilute conditions (1 μM chaperone) and in the absence of crowding agents, making it likely that the TMD dissociated prior to the assay.”

>This is not a detail, but a core result. Downplaying these strong and statistically highly significant changes as "very subtle" and "details" does not appear appropriate. The authors claim to have "clarified the text" in this respect, but it is not clear where and how. This central point needs to be addressed.

Response: To help clarify and provide additional analysis, we have calculated the statistical differences between the mutants in Figure 6a with further interpretation in the text. The statistics support that L66 is more important to TMD binding than Q58, in perfect agreement with the biophysical data.

“Furthermore, the donor strain encoding *eagT6(Q58A)* displays intermediate fitness against the recipient strain. This is consistent with our biophysical finding that Q58 contributes less significantly than L66 to the stability of the EagT6-TMD complex. The agreement between these genetic results and our biophysical data suggest that the determinants of complex stability identified *in vitro* closely reflect the function of the EagT6-TMD complex in *P. aeruginosa*.”

Also edited:

“Furthermore, the relative abundance of Tse6 in the EagT6 Q58A and L66A variant backgrounds correlates with the contribution of those residues to competitive fitness and *in vitro* TMD binding energy (Figures 4-6 and Table 1). Notably, the relative abundance of Tse6 is high for EagT6 Q58A as compared to other the mutants and wild-type (Figure 6B). Supporting this, EagT6 Q58A has a $\Delta G^{\circ}_{binding}$ of -60 kJ/mol which is still indicative of a stable complex. Regardless, in agreement with our biophysical data the loss of TMD binding affinity by Q58A significantly decreases Tse6 secretion (Figure 6A).”

Reviewer #4 (Remarks to the Author):

Response: We again thank the reviewer for their time and help to improve our study.

Reviewer #5 (Remarks to the Author):

I don't have further question.

Response: We thank the reviewer for their time and help to improve our work. We are glad we were able to address all concerns.